# Essential role for InSyn1 in dystroglycan complex integrity and cognitive behaviors in mice

**Akiyoshi Uezu[1], Erin Hisey[1], Yoshihiko Kobayashi[1], Yudong Gao[1], Tyler WA Bradshaw[1], Patrick Devlin[1], Ramona Rodriguiz[2,3], Purushothama Rao Tata[1,4], Scott Soderling[1,4]\***

[1]Department of Cell Biology, Duke University Medical School, Durham, United States; [2]Department of Psychiatry and Behavioral Sciences, Duke University Medical School, Durham, United States; [3]Mouse Behavioral and Neuroendocrine Analysis Core Facility, Duke University Medical School, Durham, United States; [4]Department of Neurobiology, Duke University Medical School, Durham, United States

**Abstract** Human mutations in the dystroglycan complex (DGC) result in not only muscular dystrophy but also cognitive impairments. However, the molecular architecture critical for the synaptic organization of the DGC in neurons remains elusive. Here, we report Inhibitory Synaptic protein 1 (InSyn1) is a critical component of the DGC whose loss alters the composition of the GABAergic synapses, excitatory/inhibitory balance in vitro and in vivo, and cognitive behavior. Association of InSyn1 with DGC subunits is required for InSyn1 synaptic localization. InSyn1 null neurons also show a significant reduction in DGC and GABA receptor distribution as well as abnormal neuronal network activity. Moreover, InSyn1 null mice exhibit elevated neuronal firing patterns in the hippocampus and deficits in fear conditioning memory. Our results support the dysregulation of the DGC at inhibitory synapses and altered neuronal network activity and specific cognitive tasks via loss of a novel component, InSyn1.

**\*For correspondence:**
scott.soderling@duke.edu

**Competing interests:** The authors declare that no competing interests exist.

## Introduction

GABA-mediated inhibitory synaptic-transmission is fundamental to almost all aspects of brain function and its dysregulation is thought to contribute to many neurological disorders including epilepsy, intellectual disability, autism and schizophrenia (*Marín, 2012*; *Zoghbi and Bear, 2012*). In the CNS, inhibitory synaptic events are mediated by GABA$_A$Rs, a heteropentamer complex assembled from a family of 19 subunits. Postsynaptic GABA$_A$Rs mediating phasic inhibition consist of two α1, α2, or α3 subunits with two β2 or β3 subunits, and a single γ2 subunit (*Olsen and Sieghart, 2008*). The spatial distribution, recruitment, and stabilization of GABA$_A$Rs at the inhibitory postsynaptic complex (iPSD) are tightly regulated by intracellular signaling molecules and trans-synaptic adhesion proteins that are still poorly understood.

The dystrophin/dystroglycan complex (DGC) is better known for its importance for the integrity of muscle fibers against mechanical stress by physically bridging the extracellular matrix and the actin cytoskeleton (*Allikian and McNally, 2007*; *O'Brien and Kunkel, 2001*; *Sunada and Campbell, 1995*). However, muscular dystrophies, which are caused by gene mutations encoding this protein complex, are often associated with cognitive deficits, epilepsy and other neurological disorders, highlighting its importance in the nervous system (*Godfrey et al., 2011*; *Hendriksen and Vles, 2008*; *Waite et al., 2012*). In neurons, DGC forms a complex with α-, β-dystroglycan, dystrophin and distinct isoforms of dystrobrevin and syntrophin. They co-localize with a subset of inhibitory synapses in distinct brain regions and are important for targeting and modulating GABAA receptors

(*Côté et al., 2002*; *Knuesel et al., 1999*). Dystroglycan and its glycosylation may be crucial for the expression of homeostatic synaptic plasticity at GABAergic synapses (*Pribiag et al., 2014*). However, the underlying mechanisms for DGC formation or stability at inhibitory synapses are unknown.

Recently, we and others developed proximity-based labeling approaches to capture the iPSD proteome in cultured neurons or directly from the mouse brain (*Loh et al., 2016*; *Uezu et al., 2016*). Our prior study successfully identified more than 180 proteins from the inhibitory post-synaptic structure. InSyn1 was one of the abundant previously uncharacterized proteins (Human: UPF0583 protein C15orf59, Mouse: 6030419C18Rik) in our iPSD proteome dataset, and we found it is functionally important for synaptic inhibition both at the single-cell and at the network level (*Uezu et al., 2016*). However, how InSyn1 functions and thus the molecular mechanism by which it regulates GABAergic inhibition was unclear.

Here, we report that InSyn1 is a crucial regulator of the dystrophin/dystroglycan complex at GABAergic synapses that is important for aspects of cognitive behavior. Cell-based structure-function and co-immunoprecipitation (co-IP) assays indicate InSyn1 interacts with multiple DGC components through its unique N-terminus. InSyn1 lacking this region no longer localizes to inhibitory synapses of neurons, indicating the interaction through DGC is crucial for its proper targeting. CRISPR-mediated depletion of αDG, but not gephyrin, disrupts InSyn1 localization, demonstrating InSyn1 requires the DGC for its proper localization. Furthermore, InSyn1 null hippocampal neurons exhibit a significant alteration in DGC and GABA$_A$R composition at inhibitory synapses. Consistent with the altered architecture of the iPSD, loss of InSyn1 results in elevated neuronal excitation with significant perturbations in neuronal bursting in vitro. Finally, InSyn1 null mice exhibit deficits in memory retrieval in a hippocampus-dependent cognitive task that is coincident with elevated c-fos immunoreactivity and a dramatic increase of neuronal activity in the dentate gyrus in vivo. Our results demonstrate InSyn1 functionally couples to the DGC and reorganize iPSD composition in a manner essential for proper neuronal network activity, the disruption of which is associated with impaired cognitive behavior.

## Results

### Localization of InSyn1 at inhibitory synapses requires its N-terminal region and is DGC dependent

Previously we demonstrated that InSyn1 was enriched in the iPSD and that it was highly co-localized with the markers of inhibitory synapses gephyrin and αDG, which both serve to scaffold and organize the iPSD (*Uezu et al., 2016*). However, InSyn1 lacked any identifiable protein domains and thus the mechanism by which it localized to the iPSD remained enigmatic. To gain insights into the localization mechanism of InSyn1 to inhibitory synapses, we used CRISPR-based depletion based on insertion-deletion (Indel) mutagenesis to deplete either gephyrin or the αDG subunit of the DGC to test whether either was required for InSyn1 iPSD localization. Primary hippocampal neurons prepared from *Lox-stop-Lox*-Cas9-P2A-GFP (Cas9 KI) mice were transduced with adeno-associated viral (AAV) expressing Cre and sgRNA against either *Gphn* or *Dag1* and fixed at P14. Depletion of endogenous protein levels of gephyrin or αDG was confirmed by immunostaining of both proteins, demonstrating the density of gephyrin or αDG clusters was decreased by 81% and 89% compared to negative control (empty sgRNA) samples (*Figure 1A*), respectively. Next, we tested whether InSyn1-HA localization was altered in either of the scaffolding protein-depleted neurons (*Figure 1B*). In control neurons, InSyn1 clusters clearly overlapped with each inhibitory post-synaptic marker such as αDG and gephyrin. CRISPR-mediated gephyrin depletion did not alter the distribution of InSyn1 puncta. However, following αDG depletion, InSyn1 dramatically diminished its clustering and was diffuse throughout the soma and dendrites (*Figure 1B*; bottom panels). The localization pattern of InSyn1 in each condition was quantified as a distribution index, a mean absolute deviation of InSyn1-HA intensity within the dendrites (*Figure 1C*). To further evaluate whether endogenous InSyn1 localization is also αDG dependent, we took advantage of Homology-Independent Targeted Integration (HITI) method to label C-terminus of InSyn1 with a highly antigenic spaghetti-monster tag (smFP-HA) in Cas9 KI neurons (*Figure 1—figure supplement 1*) (*Suzuki et al., 2016*; *Viswanathan et al., 2015*). We found a dramatic reduction of endogenous InSyn1-labeled neurons in CRISPR depleted αDG samples but no difference in gephyrin depletion compared to control (Fig. D and E). These

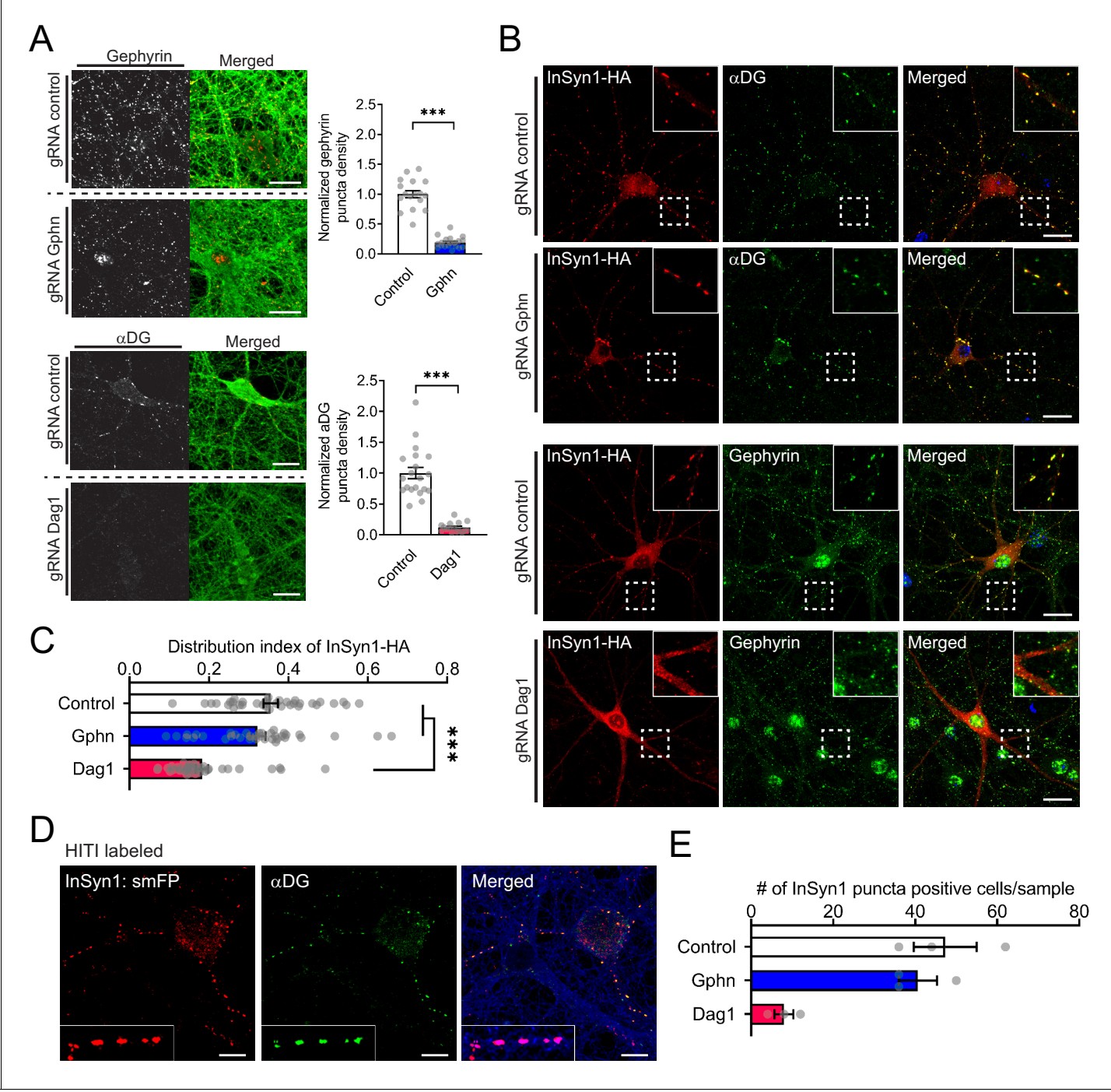

**Figure 1.** InSyn1 localization to the iPSD is DGC dependent. (A) Depletion of Gephyrin or αDG by CRISPR in neurons. Cas9 knock-in hippocampal neurons were transduced with AAV:Cre/(control)gRNA [control], AAV:Cre/(Gphn)gRNA [gephyrin] or AAV:Cre/(Dag1)gRNA [αDG] at DIV1 and stained with gephyrin or αDG at DIV13 (left panel). GFP fluorescence of the Cas9-2A-GFP (right panel). Graphs to the right show the normalized puncta density. Gphn vs control (two-tailed *t*-test, gephyrin n = 19, control n = 17, p<0.0001), Dag1 vs control (two-tailed *t*-test, αDG n = 16, control n = 20, p<0.0001). (B, C) InSyn1-HA localization after αDG or gephyrin CRISPR depletion. Neurons were depleted of αDG or gephyrin, followed by AAV:InSyn1-HA transduction 3 days before fixation. Exogenously expressed InSyn1-HA is shown in red. Endogenous gephyrin or αDG are shown in green. Bar graph showing the distribution index of InSyn1-HA as arbitrary units. Control (n = 36), Gphn (n = 36), Dag1 (n = 36). One-way ANOVA followed by Tukey's multiple comparisons test, F (2, 105)=25.49, ***p<0.001. Scale bars, 20 μm. (D). HITI labeling of endogenous InSyn1 with smFP-HA in Cas9 KI neurons. InSyn1 is shown in red and αDG is shown in green. Of note, InSyn1:smFP showed clear puncta staining colocalized with αDG. (E) InSyn1 puncta-positive cells were quantified in either Control, Gphn, or Dag1 depleted neurons (n = 3).

*Figure 1 continued on next page*

*Figure 1 continued*

The online version of this article includes the following figure supplement(s) for figure 1:

**Figure supplement 1.** Schematic illustration of HITI labeling.

quantitative analyses confirmed that αDG depletion severely disrupted the stereotypic InSyn1 localization within neurons. However, gephyrin-targeted CRISPR depletion did not alter the distribution of InSyn1 supporting the hypothesis that the inhibitory post-synaptic protein InSyn1 appears to be dependent on the DGC to manifest its synaptic localization in neurons.

InSyn1 is a 292 amino acid protein and the only potential structural insight available is that the first 60 amino acids region harbors a predicted coiled-coil sequence, which often mediates protein-protein interactions (*Burkhard et al., 2001*). To determine the regions of InSyn1 that are important for its iPSD targeting, we performed a cell-based structure-function analysis by expressing a series of truncation mutants in neurons (*Figure 2A*). AAV expressing InSyn1 truncation mutants fused to GFP were transduced into hippocampal neurons and immunostained for endogenous αDG (*Figure 2B*). Full-length InSyn1-GFP, as well as truncation mutants of the C-terminus or middle portions of the protein exhibited puncta staining that strongly colocalized with αDG. In contrast, an N-terminal truncated version, InSyn1ΔN-GFP, showed diffuse expression throughout the neurons (*Figure 2C*). Quantitative analysis, comparing the distribution index of each InSyn1 deletion mutant to soluble GFP, confirmed the loss of the N-terminal region (a.a. 1–60) disrupted its localization in neurons (*Figure 2C*).

We next sought to determine how the N-terminus of InSyn1 might mediate its co-localization with the DGC. The DGC is composed of several subunits including α1-syntrophin and dystrobrevin (*Figure 2D*). α1-syntrophin is known to function as a docking site for various proteins through its PDZ and PH domains, including ion channels, GPCRs, water channels, kinases and phosphatases in the nervous system (*Brenman et al., 1996*; *Connors et al., 2004*; *Neely et al., 2001*), while the C-terminal SU domain mediates its interaction with dystrophin and utrophin (*Kramarcy et al., 1994*; *Yang et al., 1995*). Dystrobrevin has two isoforms, α and β-dystrobrevin. Dystrobrevins are an integral component of DGC that interact with dystrophin and syntrophin as well as other signaling proteins, such as dysbindin, Kif5, and the regulatory subunit of PKA (*Benson et al., 2001*; *Ceccarini et al., 2007*; *Macioce, 2003*). Mice lacking both α and β-dystrobrevin show defects in inhibitory synaptic structure and motor functions, suggesting their important roles in organizing the DGC at the CNS (*Grady, 2006*). We thus tested whether InSyn1, as well as the truncation mutants, might interact with either of these components of the DGC, as well as gephyrin, which we previously demonstrated forms a complex with InSyn1 (*Uezu et al., 2016*). Interestingly, we found that while full-length InSyn1 bound to α1-syntrophin and β-dystrobrevin, only the N-terminus deletion of InSyn1 (InSyn1ΔN) disrupted the interaction with both α1-syntrophin and β-dystrobrevin (*Figure 2E*). We also assessed the interaction between these InSyn1 mutants and gephyrin and found none of the deletion mutants disrupted this interaction, suggesting multiple regions of InSyn1 are involved in the interactions with gephyrin and that truncation of the N-terminus did not generally disrupt all functions of InSyn1 (*Figure 2E*). These results demonstrate that InSyn1 associates with the DGC complex via its N-terminal predicted coiled-coil region and that this interaction is required for its proper localization in neurons.

## InSyn1 exhibits widespread expression throughout the brain and its loss significantly alters iPSD organization

As a prelude to analyzing the functional roles of InSyn1, we first examined its expression distribution patterns in the brain via a recently described proximity ligation in situ hybridization or PLISH (*Nagendran et al., 2018*). Probes to detect *Insyn1* mRNA were incubated with sagittal sections of adult mice to visualize regional *InSyn1* expression distribution. *Insyn1* mRNAs were detected throughout the mouse brain, with high expression in the hippocampus, olfactory bulb, cerebellum and modest expression in the cortex, thalamus, midbrain, and pons (*Figure 3A*). This expression pattern was specific as the negative control scramble probe did not exhibit any specific staining (*Figure 3A*). In the hippocampus, InSyn1 was robustly expressed in the granule cell layer of the dentate gyrus (DG) and other pyramidal cell layer regions such as CA1 (*Figure 3B*). *Insyn1* expression

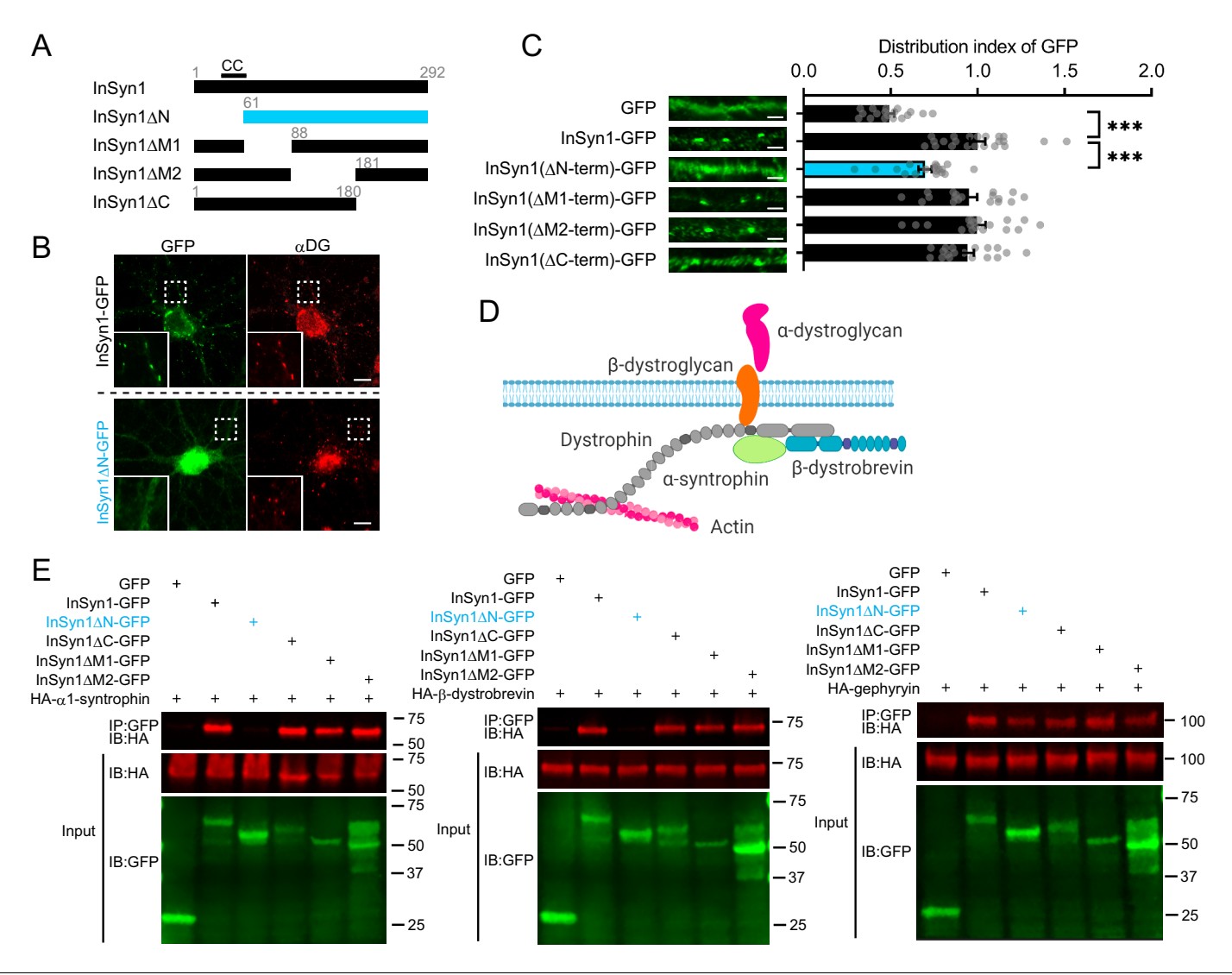

**Figure 2.** The N-terminus of InSyn1 is essential for its localization and interaction with DGC components. (A) Schematic of InSyn1 deletion mutants used for co-immunoprecipitation or localization in neurons. CC: putative coiled-coil domain. (B) Representative images of full-length or ΔN InSyn1-GFP expression (green) in neurons immunostained with antibodies to αDG (red). Scale bar; 5 μm. (C) Left; representative images of dendrites expressing each construct. Right, graphs depicting the distribution index of InSyn1 versus negative control (GFP) or InSyn1 mutants. GFP (n = 18), InSyn1-GFP (n = 22), InSyn1ΔN-GFP (n = 18), InSyn1ΔM1-GFP (n = 18), InSyn1ΔM2-GFP (n = 18), InSyn1ΔC-GFP (n = 18). One-way ANOVA followed by Tukey's post-hoc tests, F (5, 106)=23.99, ***p<0.001. Scale bars, 2 μm. (D) Schematic of the dystrophin/dystroglycan complex (DGC) in neurons. Created with BioRender.com (E) Representative immunoblots following co-immunoprecipitation of GFP (negative control), InSyn1-GFP, or InSyn1-GFP mutants with HA-α1-syntrophin, HA-β-dystrobrevin, or HA-gephyrin. Protein constructs were expressed in HEK293T cells and precipitated by GFP-Trap. IP: immunoprecipitated, IB: immunoblot.

was detected in all cells within the hilus, suggesting InSyn1 may be expressed in both excitatory and inhibitory neurons (*Pelkey et al., 2017*). In the cerebellum, we observed strong expression of *Insyn1* in Purkinje cells (PC), whereas comparatively weaker expression was detected in the internal granule cell layer (IGL).

Based on these *Insyn1* expression analysis and data demonstrating that InSyn1 requires its interaction with the DGC for its proper localization to the iPSD where it also binds gephyrin, we next examined how its loss might impact the organization of the iPSD in hippocampal neurons. To accomplish this, we first generated a new line of InSyn1 knockout mice (*InSyn1$^{-/-}$*) targeting coding sequence in exon2 of *Insyn1* gene by embryo injection of CRISPR/Cas9 and sgRNA. This resulted in

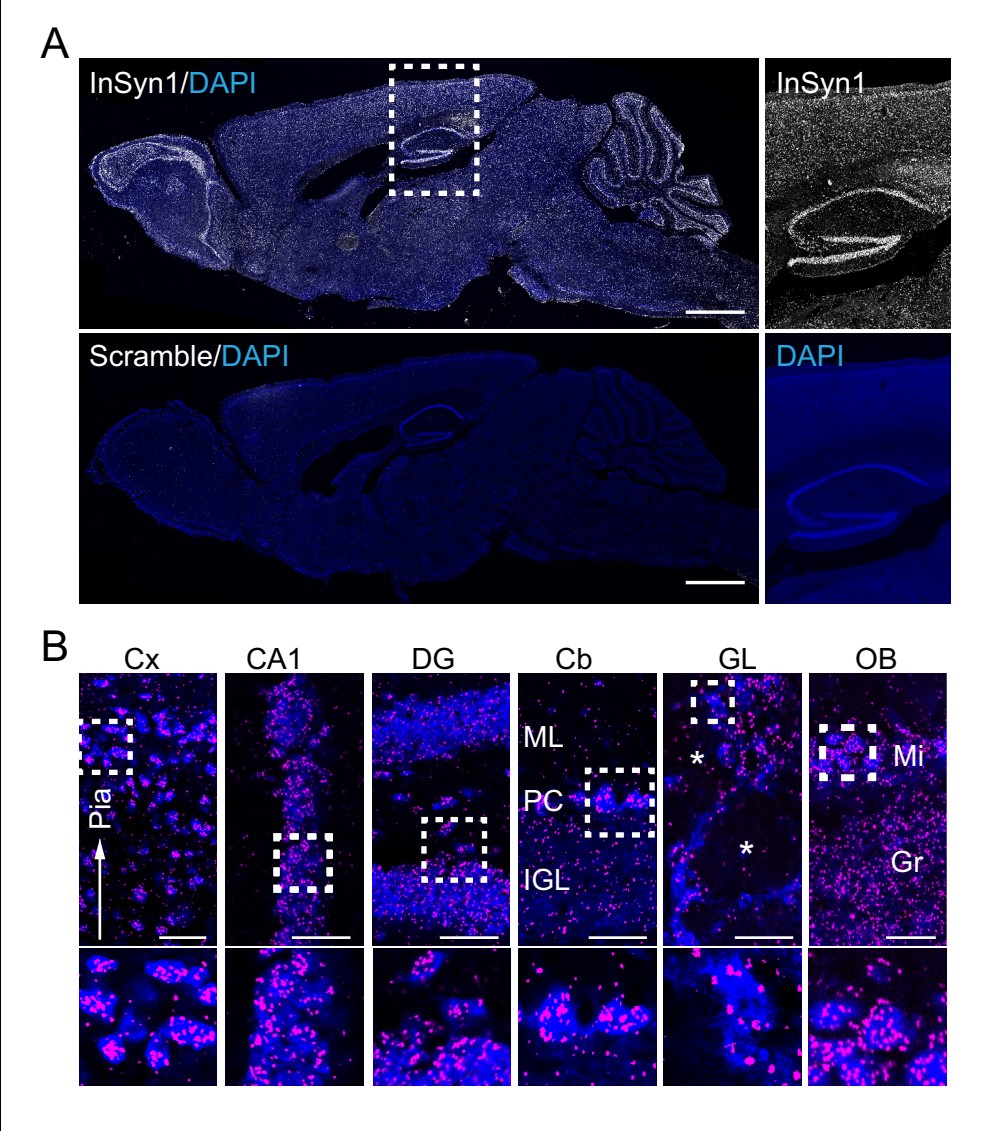

**Figure 3.** InSyn1 expression distribution in the mouse brain. (**A**) InSyn1 mRNA (white) was detected throughout the adult mouse brain with strong signals in the hippocampus, cerebellum and olfactory bulb. Nissle stain (blue). Magnified images of the hippocampus and the cortex are shown. (**B**) Numerous clusters were found in cells in different layers of the cortex (Cx), pyramidal cell layers (CA1) and dentate gyrus granule cells (DG) in the hippocampus, Purkinje cells in the cerebellum (Cb), cells surrounding the glomerulus (GL) and in the mitral cell layer of the olfactory bulb (OB). Cx; cerebrum cortex, CA1; hippocampus CA1, DG; dentate gyrus, Cb; cerebellum, OB; olfactory bulb, GL; glomerular layer, Mi; mitral cell layer, Gr; granular cell layer, ML; molecular cell layer, PCL; Purkinje cell layer, IGL; internal granule layer. The asterisk represents the glomerulus. Scale bars, 1 mm (**A**), 50 um (**B**).

The online version of this article includes the following source data and figure supplement(s) for figure 3:

**Source data 1.** Hybridization probes.
**Figure supplement 1.** Generation of InSyn1 KO mice.

mice harboring an 11-basepair missense deletion causing an out-of-frame mutation following 34 amino-acids and a premature stop codon in a highly conserved region of InSyn1 including predicted coiled-coil region (*Figure 3—figure supplement 1A–C*). In WT neurons, endogenous InSyn1 tagged with smFP-HA by HITI showed clear puncta staining on the dendritic shafts. However, InSyn1 expression was abolished in *Insyn1*[-/-] neurons confirming InSyn1 protein expression was lost following the disruptive deletion in Exon 2 (*Figure 3—figure supplement 1D and E*).

Using InSyn1 null primary hippocampal neurons, we next examined how its loss affected the organization of αDG or gephyrin by immunostaining and quantitative analysis. Loss of InSyn1 significantly impacted the density and clustering of αDG (*Figure 4A*). We found that compared to WT neurons, the density of αDG clustering was decreased by 41% in the neurite regions (*Figure 4A*). In contrast, analyzing the αDG cluster area demonstrated that there is an apparent increase in both regions (neurite regions; 30%, perisomatic regions; 18%) (*Figure 4B*). The total αDG cluster area did not show a difference between WT and InSyn1 KO neurons, suggesting the overall expression of αDG has not changed and is instead re-distributed to the fewer remaining αDG clusters in the KO neurons (*Figure 4—figure supplement 1*). Furthermore, another DGC component dystrophin showed a similar region-specific reduction of puncta density in KO neurons (*Figure 4—figure supplement 2*). Together, these data demonstrate InSyn1 regulates the spatial distribution of αDG and that its loss results in a corresponding alteration across αDG-positive synapses. We also performed the same analysis of gephyrin following the loss of InSyn1 (*Figure 4C*). Consistent with our in vitro findings, there was no significant change in the density or the cluster size of gephyrin between WT and KO neurons both in neurite and perisomatic regions (*Figure 4D*).

Because the loss of InSyn1 affects the DGC, we next examined the effects of its loss on several GABA$_A$ receptor (GABA$_A$R) subunits by immunostaining KO neurons and quantifying the density and area of GABA$_A$Rs along with Vgat, an inhibitory presynaptic marker protein at both the perisomatic or neurite regions (*Figure 5A*). In InSyn1 KO neurons, there was no significant change in the cluster density of GABA$_A$Rα1 in either region compared to WT (*Figure 5B*). We also found no significant changes in the expression of β3, which is an important subunit in inhibitory transmission in the hippocampus CA1 regions (*Nguyen and Nicoll, 2018*). However, we found a marked reduction of α2 cluster density in the perisomatic (32%) and neurite (43%) regions. The γ2 subunit is highly expressed throughout the brain and is required for GABA$_A$R clustering and normal inhibitory synaptic transmission (*Essrich et al., 1998*; *Pritchett et al., 1989*). In the perisomatic region, γ2 showed a 28% reduction of cluster density, consistent with our previous CRISPR-dependent acute depletion (*Uezu et al., 2016*). There was also a reduction in the area of α2 and β3 clusters in the neurite regions (*Figure 5B*). We have noticed that αDG staining was positive only in a subset of hippocampal neurons. Additionally, it is known that DGC localizes in a subset of the inhibitory synapses. This was verified by a previous study co-staining hippocampal neurons with GABAARα2 subunit and dystrophin (*Brünig et al., 2002*). Furthermore, it has been shown that the DGC exists only in a fraction of gephyrin-positive synapses by co-staining with βDG and gephyrin (*Lévi et al., 2002*). Based on this information, we sought to analyze only αDG negative neurons that should mainly express gephyrin. After applying the same analysis method, we found a decrease of α2 subunit cluster density in the neurite region (*Figure 5—figure supplement 1*). Together, these data show that InSyn1 has a subunit-specific effect on GABA$_A$Rs distribution in hippocampal neurons.

## Role of InSyn1 in modulating network excitability during neuronal development

We have previously shown that acute CRISPR-mediated InSyn1 depletion specifically reduced mIPSC frequency without impacting mEPSCs, which is consistent with the observations here that loss of InSyn1 leads to the abnormal distribution of αDG and GABA$_A$R α2/γ2. To broadly test how impaired inhibition may impact neuronal excitability and network activity, we measured field recordings of spontaneous and optically stimulated activity using multi-electrode arrays (MEA). Because these recordings are non-invasive, we also took advantage of repeated recordings over two weeks to assess whether InSyn1 might also modulate neuronal activity at different developmental stages in vitro (*Figure 6A*). To determine the expression of endogenous InSyn1 during neuronal development, we labeled the protein by HITI technique to image its expression course over time. InSyn1 was detected as early as DIV6 in neuronal culture and gradually increased during the development (*Figure 6—figure supplement 1*). Previous studies have shown that DGC accumulates progressively in cultured neurons prepared from rat embryos at E18 (*Lévi et al., 2002*). They found β-DG was detected from 1 week and the colocalization with GABARγ2 increased from 2 weeks of in vitro culture. These data support the functional relationship between DGC and InSyn1. Cultured cortical neurons begin with random neuronal firing, which gradually exhibits network activity characterized by increasing spike firing rate, long bursting activity, and global synchronization that indicate the maturation of neuronal connectivity (*Figure 6B*). We found an increase in the mean spontaneous and

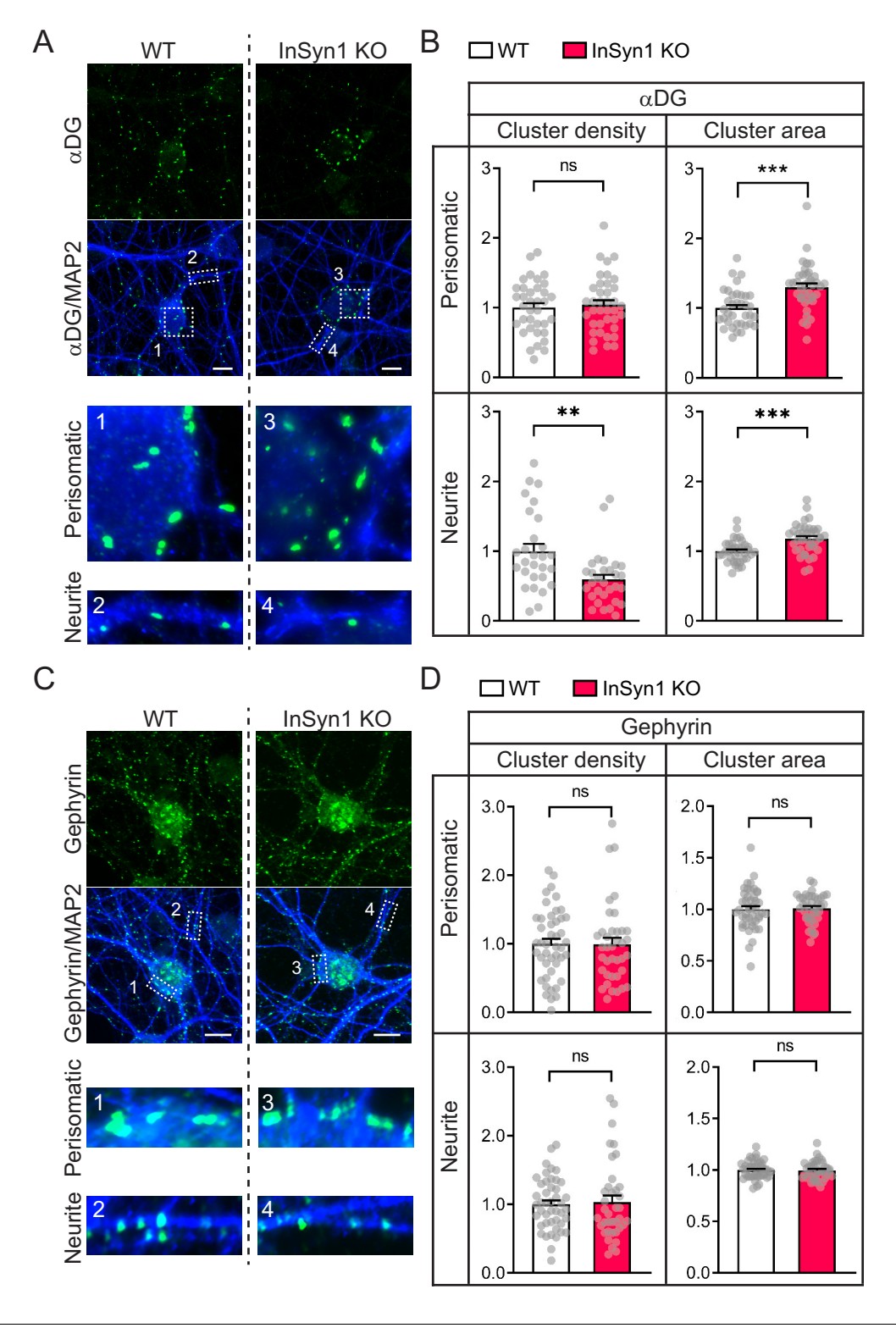

**Figure 4.** InSyn1 modulates the distribution of the DGC but not gephyrin in neurons. (**A**) Representative images of WT and InSyn1 KO hippocampal neurons at DIV13, labeled with antibodies to αDG (green) and MAP2 (blue). Scale bars, 10 um. (**B**) Bar graphs showing normalized αDG cluster density or area size either at perisomatic regions or at neurite regions. Perisomatic cluster density (WT n = 36, KO n = 38, two-tailed *t*-test, p=0.6491), neurite cluster density (WT n = 28, KO n = 30, two-tailed *t*-test, p=0.0010), perisomatic cluster area (WT n = 36, KO

*Figure 4 continued on next page*

*Figure 4 continued*

n = 38, two-tailed *t*-test, p<0.0001), and neurite cluster area (WT n = 36, KO n = 38, two-tailed *t*-test, p<0.0001).
*p<0.05, **p<0.001, ***p<0.0001. (C) Representative images of WT and InSyn1 KO hippocampal neurons labeled
with antibodies to gephyrin (green) and MAP2 (blue). Scale bars, 10 μm. (D) Bar graphs showing normalized
gephyrin cluster density or area size. Perisomatic cluster density (WT n = 45, KO n = 38, two-tailed *t*-test,
p=0.5963), neurite cluster density (WT n = 45, KO n = 38, two-tailed *t*-test, p=0.6331), perisomatic cluster area (WT
n = 45, KO n = 39, two-tailed *t*-test, p=0.7923), and neurite cluster area (WT n = 36, KO n = 38, two-tailed *t*-test,
p=0.8098).

The online version of this article includes the following figure supplement(s) for figure 4:

**Figure supplement 1.** Quantification of αDG area in hippocampal neurons.
**Figure supplement 2.** InSyn1 modulates the distribution of dystrophin in neurons.

evoked firing rate of InSyn1 null neurons compared to WT at DIV8 and DIV11, but no difference at DIV14 (*Figure 6—figure supplement 2A*) as well as no differences in network synchrony (*Figure 6—figure supplement 2B and C*). These effects on firing rates corresponded well with a significant increase in the frequency of neuronal bursting at early time points (DIV8; 26%, DIV11; 20%) (*Figure 6C*). Interestingly, at DIV14 the frequency of neuronal bursts was no longer different between WT and InSyn1 null neurons. However, the duration of neuronal bursts (DIV14) and the number of spikes per burst (DIV11 and 14) were significantly increased at later time points (*Figure 6C*). Bursting activity in cultured neuronal networks critically depends on excitatory synaptic transmission while GABAergic inputs participate in the termination of the bursts (*Cohen et al., 2008*; *Suresh et al., 2016*). We also found an increase in the number of spikes per burst at the network level at an early time point as well as an increase in the frequency of synchronous burst at the later developmental stage (*Figure 6D*). These data demonstrated that InSyn1 depletion increases the bursting activity and the duration of neuronal networks in culture, presumably originating from the inhibitory synaptic defects we previously identified.

We further characterized the functional consequences of InSyn1 on network activity using optogenetic modulation. Optogenetic tools allow precise control of spike timing using short pulses of blue light to excite light-gated ion channels such as channelrhodopsin-2 (ChR2) (*Boyden et al., 2005*; *Gunaydin et al., 2010*). We measured the spike count and the latency to the first spike in response to the optical stimulation from neurons transduced with AAV-hSyn-hChR2(H134R)-EYFP. Neurons were infected at DIV1 and evoked spike activities were recorded with a range of different light intensities at DIV8, 11, and 14 (*Figure 6—figure supplement 2D*). At DIV8, InSyn1 null neurons exhibited a significant increase in the spike count at the stimulation intensity of 75%. At DIV11, InSyn1 null neurons still exhibited an increase response, however at DIV14, there was no statistical difference between the two genotypes (*Figure 6—figure supplement 2E*). These effects were not likely a reflection of altered membrane properties or channel kinetics as the first spike latency, which is a decay time of the first spike response to the stimulus, showed no genotype effects at any of the developmental stages (*Figure 6—figure supplement 2F*).

## Behavioral analysis of InSyn1 KO mice

The above data strongly supports the contention that InSyn1 is associated with and functionally important for aspects of neuronal and network bursting properties as well as the spatial distribution of GABA$_A$R subtypes and the inhibitory DGC. Interestingly, muscular dystrophies arising from mutations in the DGC are often associated with mild to severe cognitive defects, epilepsy, and other neurological abnormalities (*Hendriksen and Vles, 2008*). These phenotypes are thought to arise due to the roles of the DGC at GABAergic synapses as abnormalities in inhibitory synaptic transmission are known to contribute to various neurological disorders, including epilepsy (*Möhler, 2006*). To determine whether InSyn1 might have similar physiologic functions, we performed a range of tests to determine if the loss of InSyn1 leads to increase sensitivity to seizure induction, altered locomotor activity levels, anxiety, or abnormalities in measures of cognition.

We first examined whether InSyn1 KO mice are prone to seizures. Flurothyl is a GABA receptor antagonist that can provide a reliable measurement of seizure threshold due to altered inhibition (*Judson et al., 2016*; *Krasowski, 2000*). We measured the latency to induction of myoclonic seizures, assessed as a brief contraction of the body musculature and generalized seizures as the mouse

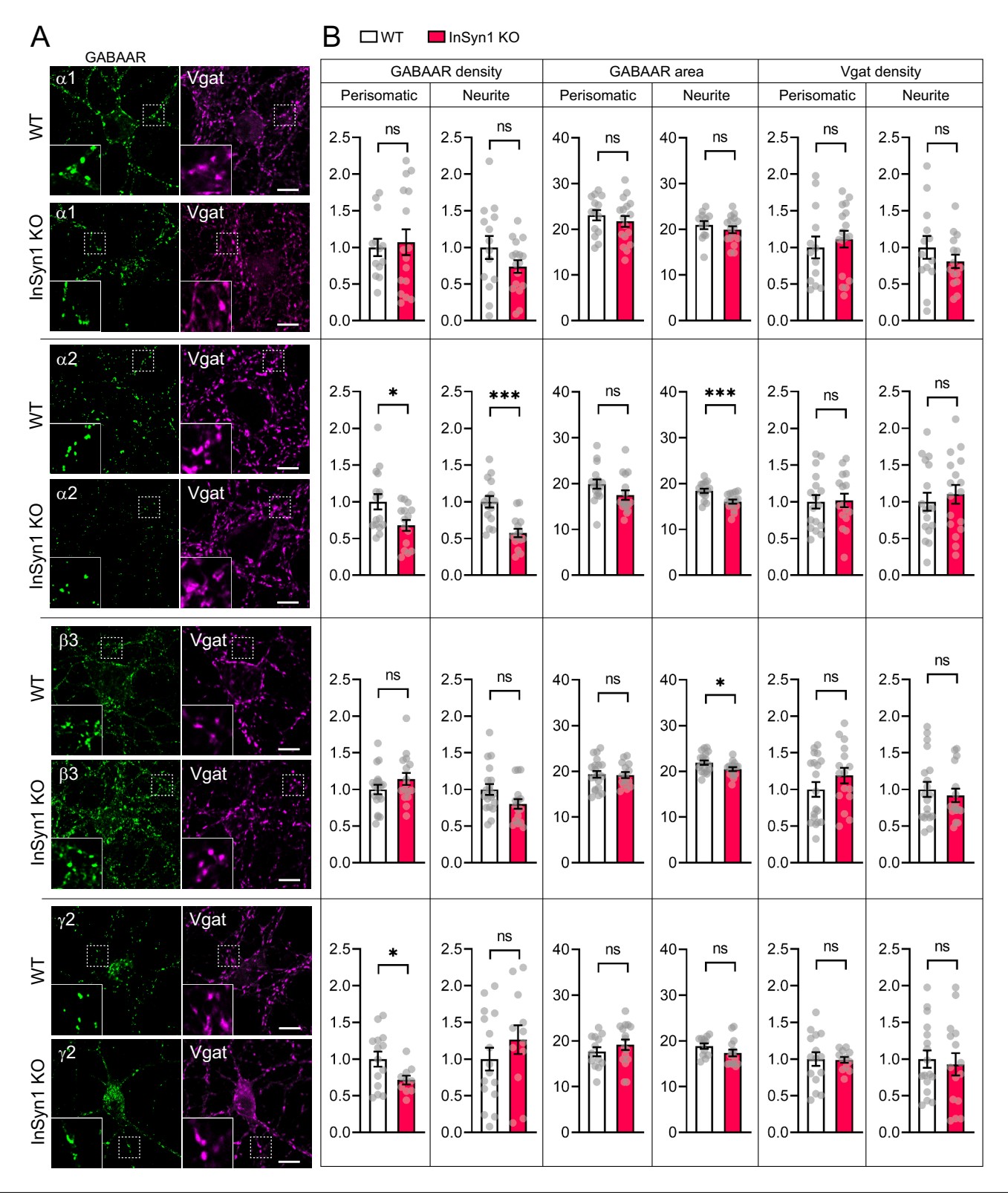

**Figure 5.** Altered distribution of GABA$_A$R clustering in InSyn1 KO neurons. (**A**) Representative images of WT and InSyn1 KO neurons at DIV13, immunostained with α1, α2, β3, and γ2 GABA$_A$R subunits (green), and Vgat (magenta). Scale bars, 10 μm. (**B**) Bar graphs showing the quantification of normalized cluster density and area size of each GABA$_A$R and Vgat from perisomatic or neurite regions. GABA$_A$Rα1 density (Two-tailed *t* test, Perisomatic; WT n = 13, KO n = 16, p=0.7421. Neurite; WT n = 14, KO n = 17, p=0.1589), and area (Two-tailed *t* test, Perisomatic; WT n = 14, KO

*Figure 5 continued on next page*

Figure 5 continued

n = 17, p=0.4237. Neurite; WT n = 12, KO n = 16, p=0.4012). GABA$_A$Rα2 density (Two-tailed $t$ test, Perisomatic; WT n = 16, KO n = 15, p=0.0179. Neurite; WT n = 17, KO n = 18, p=0.0021), and area size (Two-tailed $t$ test, Perisomatic; WT n = 16, KO n = 16, p=0.1083. Neurite; WT n = 16, KO n = 16, p=0.0009). GABA$_A$Rβ3 density (Two-tailed $t$ test, Perisomatic; WT n = 19, KO n = 16, p=0.1874. Neurite; WT n = 17, KO n = 18, p=0.0522), and area size (Two-tailed $t$ test, Perisomatic; WT n = 20, KO n = 15, p=0.8574. Neurite; WT n = 19, KO n = 15, p=0.0353). GABA$_A$Rγ2 density (Two-tailed $t$ test, Perisomatic; WT n = 14, KO n = 10, p=0.0254. Neurite; WT n = 16, KO n = 12, p=0.2996), and area size (Two-tailed $t$ test, Perisomatic; WT n = 13, KO n = 16, p=0.3262. Neurite; WT n = 12, KO n = 15, p=0.1271). Of note, no significant difference was found in Vgat cluster quantifications (Two-tailed t-test. GABA$_A$Rα1, Perisomatic; WT n = 13, KO n = 17, p=0.5510. Neurite; WT n = 13, KO n = 15, p=0.2996. GABA$_A$Rα2, Perisomatic; WT n = 17, KO n = 16, p=0.8858. Neurite; WT n = 17, KO n = 17, p=0.5657. GABA$_A$Rβ3, Perisomatic; WT n = 19, KO n = 16, p=0.3011. Neurite; WT n = 19, KO n = 16, p=0.5536. GABA$_A$Rγ2, Perisomatic; WT n = 15, KO n = 12, p=0.9152. Neurite; WT n = 17, KO n = 15, p=0.7124). *p<0.05, ***p<0.001.

The online version of this article includes the following figure supplement(s) for figure 5:

**Figure supplement 1.** Quantification of GABA$_A$Rs in αDG-negative cells.

lost the posture. However, there was no difference between WT and InSyn1 KO mice (*Figure 6—figure supplement 3*). We next performed two different tasks to evaluate their cognitive functions. Novel object recognition examines nonspatial hippocampal-dependent learning and memory (*Roberts et al., 2009*). Both short- and long-term memories were intact in KO mice as they were able to distinguish the familiar and the novel object at 30 min and 24 hr following the training (*Figure 6—figure supplement 4*). The Morris water maze is a test for spatial learning and memory (*Figure 6—figure supplement 5A*). In this test, mice were trained to escape from a pool of water by swimming to a hidden platform using the special cues. During 8 days of learning sessions (acquisition phase), WT and KO mice spent a similar time to reach the platform. A similar result was found when the platform was relocated to the opposite quadrant (reverse acquisition phase) (*Figure 6—figure supplement 5B*). Only during the reversal probe trials, KO mice showed a small potential improvement at DIV10 compared to WT littermates (*Figure 6—figure supplement 5C*). Overall, the data suggest InSyn1 KO mice have normal spatial navigational learning capabilities.

We next analyzed locomotor activity and anxiety levels of the InSyn1 KO mice in the open field. InSyn1 KO mice exhibited a slight but significant decrease in total distance moved (*Figure 7A–B*). However, no significant changes were observed in stereotypical behavior or time spent in the center of the arena (*Figure 7C–D*). These data indicated InSyn1 KO mice are hypoactive compared to their WT littermates but do not exhibit alterations in stereotypy or anxiety. Analysis of InSyn1 KO behavior in the light-dark box paradigm confirmed these observations, with no differences in the latency to cross from a darkened to a lighted box, number of transitions, or time spent in the light (*Figure 7E–G*). Again, the total activity as a measure of the number of beam brakes was reduced, verifying InSyn1 KO mice exhibit hypoactivity (*Figure 7H*).

Based on our findings that InSyn1 mRNA expression is abundant in the hippocampus and that its deletion resulted in both altered DGC and GABA$_A$R distribution as well as elevated neuronal excitability in vitro, we next tested whether InSyn1 null mice showed abnormal hippocampal activity during exploration of the open field in vivo. To analyze in vivo neural activity, we infected neurons in the dentate gyrus of the dorsal hippocampus with GCaMP6f-expressing AAV under the control of CaMKII promoter, followed by GRINS lens implantation 3–4 weeks later (*Chen et al., 2013*). In vivo imaging via a head-mounted miniaturized 1P microscope was utilized to image and record neural activity via Ca$^{2+}$-induced fluorescence events in both WT and InSyn1 null mice during exploration (*Figure 8A and B*)(*Ghosh et al., 2011*). Prior electrophysiological and calcium imaging studies report a sparse and a low-frequency neuronal activity in which the rate depends on the status of animal behavior in this task (*Danielson et al., 2016*; *Leutgeb et al., 2007*; *Pernía-Andrade and Jonas, 2014*). Raster plots of calcium events depict relatively sparse neuronal activity in both WT and KO animals during exploration of the open field (*Figure 8B*). Quantification of calcium events revealed the event rate of KO mice was significantly elevated (WT = 0.044 Hz vs KO = 0.098 Hz), and this increase was found during both the movement (WT = 0.050 Hz vs KO = 0.121 Hz) and the stationary (WT = 0.041 Hz vs KO = 0.075 Hz) periods (*Figure 8C*). The ratio of neuronal firing events during movement versus stationary periods showed no statistical difference between genotypes (*Figure 8D–E*). Taken together, these data demonstrate that loss of InSyn1 has a profound impact on neural activity in vivo, consistent with our observations of activity in vitro.

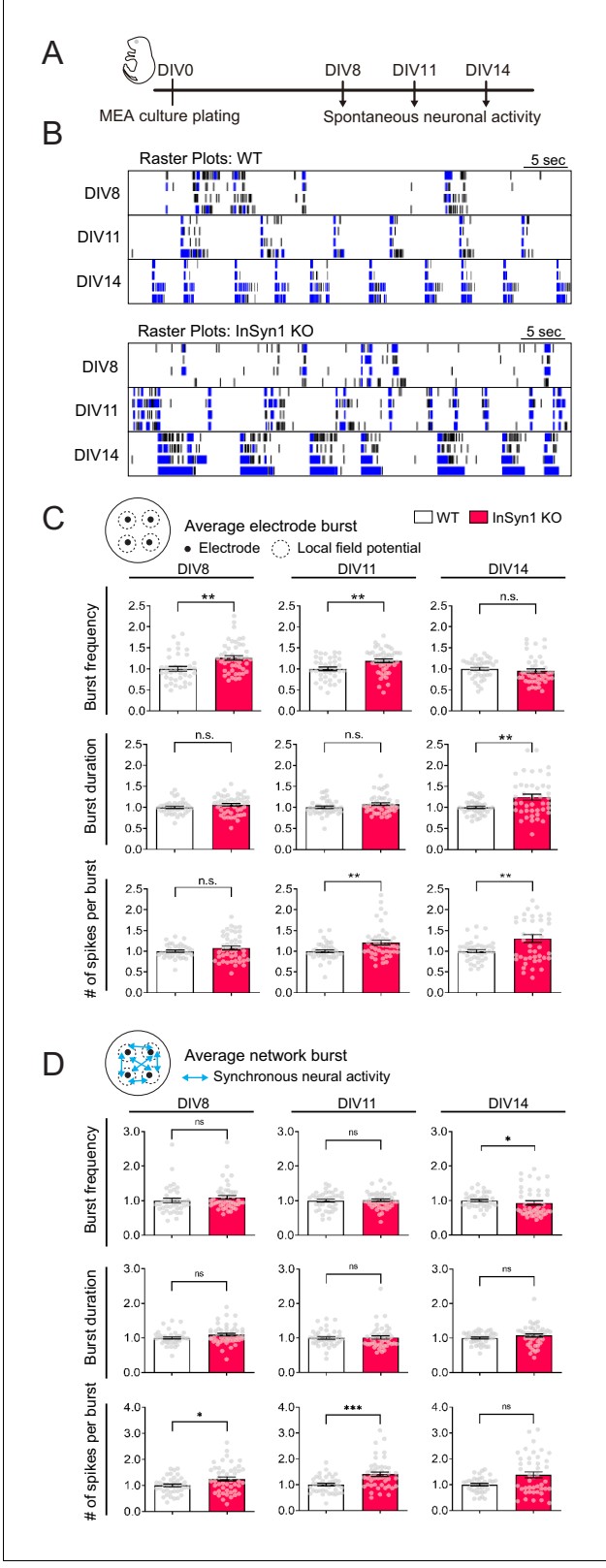

**Figure 6.** MEA recordings from InSyn1 KO neurons exhibit increased neuronal network activity. (**A**) Experimental design to record spontaneous neuronal activity. Cultured cortical neurons were prepared from InSyn1 WT or KO neonatal pups at P0, and spontaneous neuronal activities were recorded at DIV8, 11, and 14. (**B**) Representative raster plots of four electrodes from each genotype recorded at DIV8, 11 and 14. Black ticks indicate the time of a

*Figure 6 continued on next page*

*Figure 6 continued*

spike occurred, and blue ticks indicate the spikes are part of single-electrode burst activity. (**C**) KO neurons showed increased spontaneous activity compare to WT at early time points. Bar graphs showing normalized average of burst frequency (Two-tailed t-test. DIV8, WT n = 38, KO n = 46, p=0.0010. DIV11, WT n = 38, KO n = 45, p=0.0026. DIV14, WT n = 39, KO n = 43, p=0.1064), normalized average of burst duration (Two-tailed t-test. DIV8, WT n = 38, KO n = 46, p=0.1689. DIV11, WT n = 38, KO n = 45, p=0.1195. DIV14, WT n = 39, KO n = 43, p=0.0021), and normalized average of number of spikes per burst (Two-tailed t-test. DIV8, WT n = 38, KO n = 46, p=0.1460. DIV11, WT n = 38, KO n = 45, p=0.0093. DIV14, WT n = 39, KO n = 43, p=0.0038). **p<0.001. (**D**) Measurements of synchronous neuronal activities. Bar graphs showing the normalized average of network burst frequency (Two-tailed t-test. DIV8, WT n = 38, KO n = 46, p=0.1253. DIV11, WT n = 38, KO n = 45, p=0.8531. DIV14, WT n = 39, KO n = 43 from three plates, p=0.0395.), normalized average of network burst duration (Two-tailed t-test. DIV8, WT n = 38, KO n = 46, p=0.0742. DIV11, WT n = 38, KO n = 45, p=0.7676. DIV14, WT n = 39, KO n = 43, p=0.1380), and normalized average number of spikes per network burst (Two-tailed t-test. DIV8, WT n = 38, KO n = 46, p=0.0104. DIV11, WT n = 38, KO n = 45, p=0.0003. DIV14, WT n = 39, KO n = 43, p=0.0745). *p<0.05. ***p<0.001.

The online version of this article includes the following figure supplement(s) for figure 6:

**Figure supplement 1.** Developmental expression of InSyn1 in hippocampal neurons.
**Figure supplement 2.** InSyn1 KO neurons showed increased network activities but no change at the network synchrony recorded from MEA.
**Figure supplement 3.** Flurothyl-induced seizure test showed no difference between WT and InSyn1 KO mice.
**Figure supplement 4.** InSyn1- /- mice showed normal memory in novel object recognition test.
**Figure supplement 5.** Spatial memory during the Morris Water Maze learning test of InSyn1- /- mice.

---

The hippocampus plays a pivotal role in the processing of emotional information through circuitry connections with other brain regions such as the amygdala (*Engin and Treit, 2007*). This cognitive processing may involve hippocampal DGC-positive inhibitory synapses, as mice lacking dystrophin (*Mdx* null mice) exhibit altered hippocampal function and significant cognitive disturbances in fear conditioning (*Chaussenot et al., 2015*; *Dallérac et al., 2011*; *Vaillend and Chaussenot, 2017*; *Vaillend et al., 2010*). Based on these prior studies of DGC function in the CNS and our observations of the altered hippocampal activity of the InSyn1 null mice, we next examined the contextual memory capability in the fear conditioning paradigm in which mice learn to associate aversive events (mild electric shock) within a specific context (*Figure 9A*). The freezing behavior during the context test attribute to hippocampal or temporal lobe processes (*Phillips and LeDoux, 1992*). Twenty-four hours after the foot shock, InSyn1 KO mice showed a marked decrease in conditioned freezing to the context compared to their WT littermates (*Figure 9B and C*). The freezing rate immediately after the footshock, which is an index of stress-induced fear response, were comparable between WT and KO mice. (*Figure 9D*). We also evaluated the dose-response relationship to electric foot shock and found InSyn1 KO mice responses were comparable to that of WT mice (*Figure 9D*). These data demonstrate that although InSyn1 KO mice perceive sensory footshock, contextual memory of these events is significantly impaired. These effects are unlikely to be due to differences in innate anxiety levels as baseline anxiety was normal in open field and light-dark emergence testing. Instead, the effects on cognitive performance were likely downstream of the elevated activity we detected in the dentate gyrus. To further test this possibility, we next investigated whether differential activation of neurons could be observed in the hippocampus using an independent method. To this end, we quantified c-Fos expression in the DG of both phenotypes. c-Fos is an early immediate gene whose expression is a marker of neural activity in hippocampal neurons during fear conditioning and memory retrieval (*Milanovic et al., 1998*; *Reijmers et al., 2007*). Following contextual fear memory retrieval, mice were perfused and stained with a c-Fos-specific antibody (*Figure 9E*). The quantitative comparison revealed a 26% increase of c-Fos-positive cells in the DG of InSyn1 KO mice, indicative of an effect of genotype on neuronal activity in this paradigm (*Figure 9F*). In summary, these data indicate InSyn1 is vital for normal neuronal activity in the hippocampus and that its loss significantly increases neuronal activity and impairs contextual fear memory.

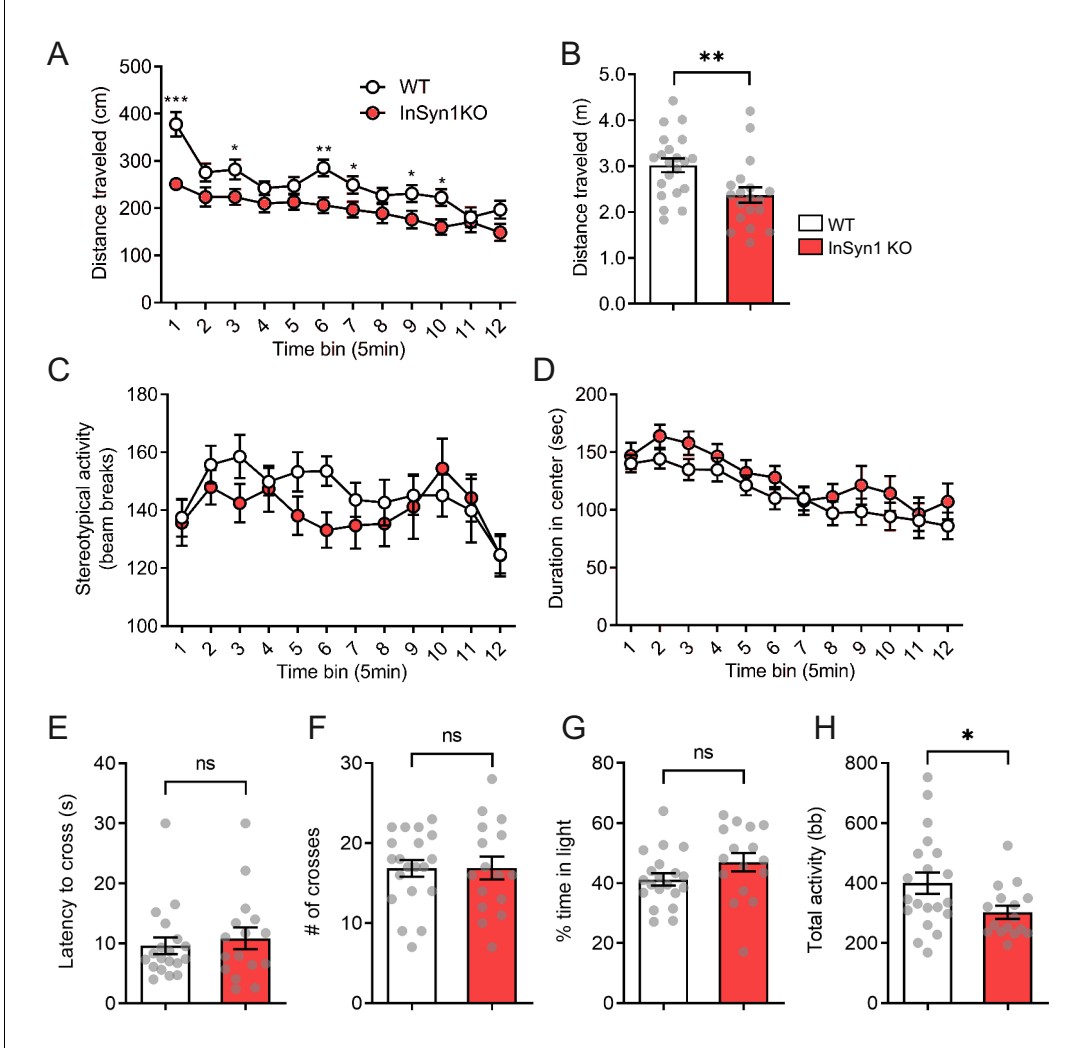

**Figure 7.** *InSyn1- /-* mice exhibit hypoactivity. Analysis of open field exploration behavior (**A–D**). (**A**) Distance traveled in the open field over 1 hr shown in 5 min time blocks (ANOVA with repeated measure, WT n = 21, KO n = 21, main effects of group $F_{(1, 38)}$=8.359, p=0.06, post-hoc t-test). (**B**) The plot of the total distance traveled (Two-tailed t-test, p=0.0066). (**C**) Graph of stereotypy behavior (ANOVA with repeated measure, WT n = 21, KO n = 21, no group difference for genotype F(1, 38)=1.186, p=0.283). (**D**) The plot of the duration spent in the center of the field (ANOVA with repeated measure, WT n = 21, KO n = 21, no group difference for genotype F(1, 38)=1.314, p=0.259). (**E–F**). Graphs depicting data from the light-dark transition test for: (**E**) Measurement of the latency to enter the light area (Two-tailed t-test, WT n = 19, KO n = 16, p=0.5970); (**F**) Total crosses between the areas (Two-tailed t-test, WT n = 20, KO n = 16, p=0.9888); (**G**) Time spent in the light area (Two-tailed t-test, WT n = 20, KO n = 16, p=0.1295); and (**H**) Total activity (Two-tailed t-test, WT n = 20, KO n = 16, p=0.0265). bb; beam break. *p<0.05, **p<0.01, ***p<0.001.

## Discussion

Recently, studies have revealed the identity of the molecular machinery of the inhibitory postsynaptic density (iPSD) using a variety of novel proteomic approaches (*Davenport et al., 2017*; *Ge et al., 2018*; *Heller et al., 2012*; *Kang et al., 2014*; *Loh et al., 2016*; *Nakamura et al., 2016*; *Tanabe et al., 2017*; *Uezu et al., 2016*; *Yamasaki et al., 2017*). The emerging picture is that the iPSD is far more molecularly complex than previously appreciated. This new framework for understanding iPSD composition highlights a pressing need to mechanistically untangle how the proteome of the iPSD governs aspects of inhibition and neuronal network activities relevant to behavior. To address this, we report here the neuronal functions of InSyn1, an uncharacterized protein previously discovered at the iPSD in our in vivo chemico-genetic proximity labeling study (*Uezu et al., 2016*).

Initially, we identified InSyn1 by its proximity to gephyrin, however, we also found InSyn1 was associated with the DGC by co-IP (*Uezu et al., 2016*). Gephyrin is a well-known scaffolding protein

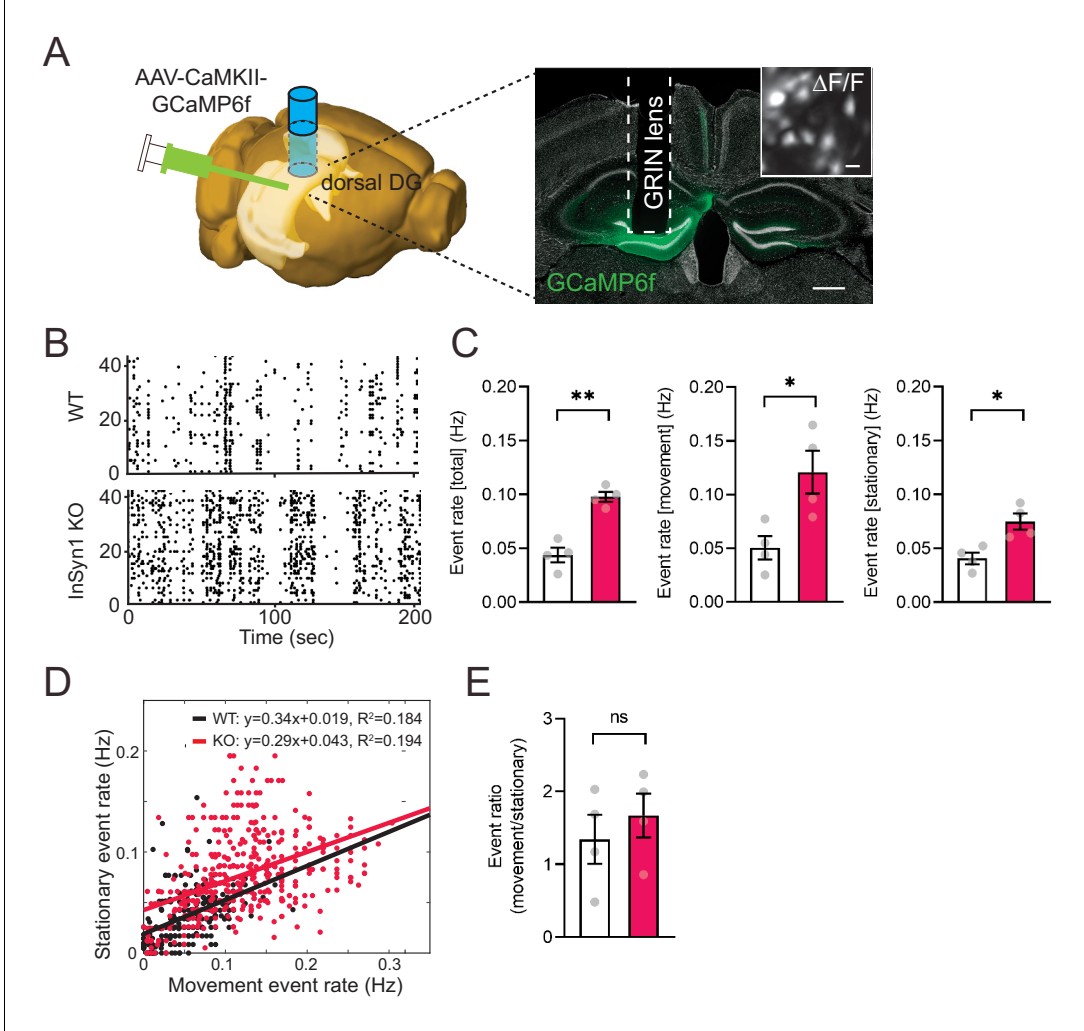

**Figure 8.** *InSyn1-/-* mice exhibit elevated neuronal activity in the dentate gyrus. (**A**) Left; experimental design showing unilateral AAV-CaMKII-GCaMP6f injection into the DG of the dorsal hippocampus. The image was generated by the Allen Institute Brain Explorer two software (http://mouse.brain-map. org/static/brainexplorer). Right; an image of GCaMP6f expression and the position of the implanted GRIN lens. Scale bar; 500 um. A ΔF/F transformed image is inserted. Scale bar; 20 μm. (**B**) Representative images of a $Ca^{2+}$-transient-raster plot from 40 cells. (**C**) Measurements of $Ca^{2+}$ event rate. (Two-tailed *t*-test, WT n = 4, KO n = 4, total, p=0.0011. movement, p=0.0269. stationary, p=0.0143). *p<0.05, **p<0.01. (**D**) Scatter plot of movement and stationary state-related $Ca^{2+}$-transient frequency recorded from WT and InSyn1 KO mice. (**E**) Bar graph showing the $Ca^{2+}$-transient event ratio of movement to stationary (Two-tailed *t*-test, WT n = 4, KO n = 4, p=0.496).

residing at the iPSD that serves as a central hub linking synaptic adhesion proteins, $GABA_A$ receptors, and signaling molecules (*Tyagarajan and Fritschy, 2014*). In the nervous system, the DGC is also considered as a scaffolding complex to organize multiple inhibitory synaptic proteins such as $GABA_A$Rs, NL2, and neurexin (*Pribiag et al., 2014*; *Sugita et al., 2001*; *Sumita et al., 2007*). Thus, one of the first questions we addressed was whether gephyrin and/or DGC serves as a scaffold for InSyn1 localization in neurons. We found CRISPR-dependent depletion of dystroglycan, but not gephyrin, disrupted the inhibitory synapse localization of InSyn1, strongly supporting the contention that the DGC is responsible for proper localization of InSyn1. As a previous study has shown the DGC co-IPs with gephyrin from rat cortical lysate, we speculate that InSyn1 may help to bridge interactions between these two scaffold proteins (*Pribiag et al., 2014*). Further, structure-function assays utilizing InSyn1 deletion mutants supported the notion that InSyn1 localization is DGC dependent.

Coiled-coil (CC) motif is a well-characterized structure consist of two right-handed a-helices wrapped around one another and is found in many protein complexes including DGC, severing as a hub to interconnect the DGC components. (*Sadoulet-Puccio et al., 1997*). C terminal domain of

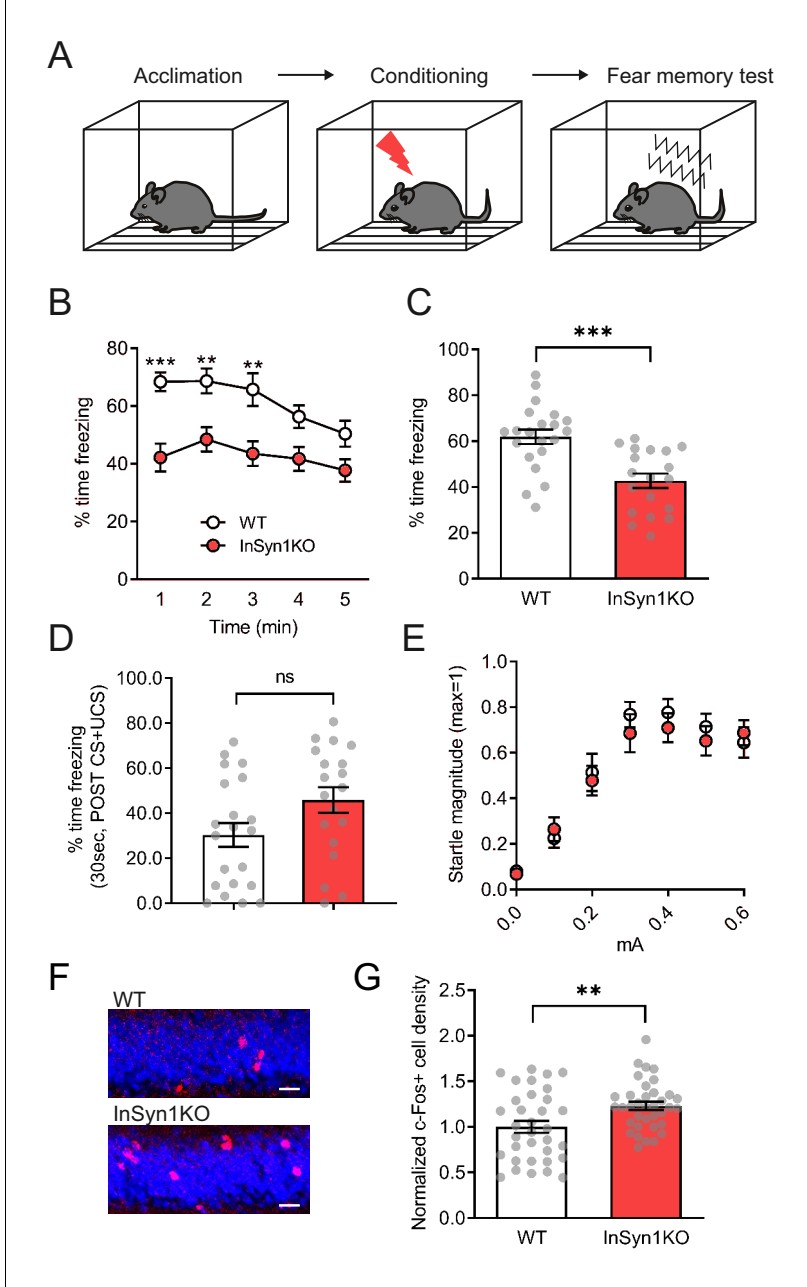

**Figure 9.** *InSyn1- /-* mice exhibit reduced memory recall and elevated c-Fos staining in contextual fear conditioning. (**A**) Experimental scheme of contextual fear conditioning. After acclimation, mice receive a mild aversive foot-shock in a conditioning chamber. The next day, freezing upon placement in the chamber (without shock) was assessed. (**B**) Graph of startle responses to foot shock. Mice were exposed to different intensities of foot shock and the vertical acceleration was quantified and normalized. Two-way repeated measure ANOVA, Genotype effect, F (1, 29)=0.7095, p=0.4065. (**C**) Graph of the freezing response after shock stimulation between WT and KO mice. (**D**) Line graph representing the time of freezing in 1 min time bins during the fear memory test. Two-way repeated measure ANOVA, Genotype effect, F(1, 38)=18.38, p=0.0001. Bonferroni post-hoc analysis, 1 min p=0.0004, 2 min p=0.0085, 3 min p=0.0172, 4 min p=0.0735, 5 min p=0.1928. *p<0.05, **p<0.01, ***p<0.001. (**E**) Graph showing the total time of freezing is reduced more than 30% in InSyn1 KO mice compared to the WT littermates (WT n = 21, KO n = 19, two-tailed t-test, p<0.0001). (**F**) Representative images of hippocampus DG from WT and InSyn1KO stained for c-Fos (red) and Nissl (blue). Scale bars; 20 μm. (**G**) Bar graph represents c-Fos positive cell density in the DG regions of WT and KO hippocampal tissues. (Two-tailed t-test, WT n = 34 from four brains, KO n = 36 from five brains, p=0.0078). **p<0.01, ***p<0.001.

dystrophin consists of a syntrophin binding domain and a CC motif. This CC motif has been shown to interact with the CC motif at the C terminal region of dystrobrevin. InSyn1 does not appear to harbor any domains of known function except a potential CC motif at the N terminus. The deletion of this CC motif disrupts InSyn1 localization at the inhibitory synapse. Additionally, we found InSyn1 interacted with α1-syntrophin or β-dystrobrevin through this conserved CC motif-containing N-terminal region. Together, these data indicate that interactions with the DGC through α1-syntrophin and/ or β -dystrobrevin is necessary for proper InSyn1 iPSD localization. Further structure-function or in vitro experiments with InSyn1 mutants will be necessary to elucidate the core region involved in protein-protein interaction with other DGC proteins. Recently, a dystrophin-EGFP reporter mouse has been generated (*Petkova et al., 2016*). Immunoprecipitate the dystrophin complex from this reporter mouse brain in combination with biochemical synaptosome preparation and mass spectrometry analysis will be an interesting avenue to reveal a comprehensive picture of the DGC molecular structure.

Little is known about the molecular mechanisms by which the DGC is selectively organized at inhibitory synapses. Utilizing InSyn1 KO hippocampal neurons, we found a specific loss of DGC clusters in dendritic regions and an increase in the size of DGC clusters in the perisomatic regions. These results demonstrate that InSyn1 and the DGC are tightly coupled to one another, with both being important to localize the other at inhibitory post-synaptic sites. While it is still not clear exactly how InSyn1 modulates DGC localization, our previous InSyn1 in vivo BioID and IP-MS studies identified several motor proteins such as the kinesins Kif3a/b, Kif2a, and Kif5a/b (*Uezu et al., 2016*). KIFs have been known to support intracellular transport of inhibitory synaptic cargos/receptors enriched in the dendrites. For example, KIF5 is involved in the transportation of the GABAA receptor or inhibitory synaptic organizer FGF7 (*Nakajima et al., 2012*; *Terauchi et al., 2015*; *Twelvetrees et al., 2010*). Additionally, it is reported that β-dystrobrevin interacts with Kif5a in the mouse brain (*Macioce, 2003*), further implying kinesins may orchestrate DGC localization at the iPSD. Thus, it is possible that InSyn1 and DGC cooperate to leverage kinesin MT-dependent transportation to synaptic sites.

We also found alterations in the composition of synaptic GABA$_A$R subunits upon loss of InSyn1. There was a decrease of GABA$_A$Rα2 clusters both at the perisomatic and dendrite regions in the KO neurons, while α1 subunit clusters were comparable between genotypes. Changes in the cluster area were also found in the GABA$_A$R α1 and α2 subunits. These results do not completely correlate with αDG cluster analysis. Currently, a detailed mechanism of how the DGC regulates inhibitory synapse function or GABA$_A$R composition/expression is not known. Using InSyn1 as a probe, our previous in vivo BioID and in vivo IP-MS studies captured several inhibitory synaptic proteins such as collybistin, Iqsec3, and GABA$_A$R subunits α1, α2 in addition to DGC components. These InSyn1 interaction partners might participate in GABA$_A$R defects we have observed in InSyn1 null neurons. In the cortex and hippocampus, two types of GABAergic basket interneurons such as PV-positive and CCK-positive cells reside. While PV+ basket cells synapses onto principal neurons mainly through α1 containing GABA$_A$Rs, inhibitory post-synapses receiving CCK+ cells are α2 subunit enriched that mediate the anxiolytic effects of benzodiazepines (*Freund and Katona, 2007*). As InSyn1 depletion mainly affected α2 subunit distribution, InSyn1 may regulate the function of GABAergic synapses receiving inputs from CCK+ interneurons. This possibility is in line with a recent study showing that CCK+ GABAergic innervation was specifically disrupted after dystroglycan was ablated from pyramidal neurons (*Früh et al., 2016*). Interestingly, depletion of Neuroplastin-65 (Np-65), a potential GABA$_A$R binding protein, also exhibited a specific α2 GABA$_A$R subunit deficiency in neurons (*Herrera-Molina et al., 2014*; *Sarto-Jackson et al., 2012*). As Np-65 was also found in our previous iPSD in vivo BioID experiment, it will be interesting to investigate potential interactions between InSyn1 and neuroplastin-65 and whether a common mechanism exists between these two proteins to regulate GABA$_A$Rs subunit expression and composition.

Previously, we have shown that acute CRISPR-dependent depletion of InSyn1 in hippocampus slice cultures exhibits inhibitory synaptic deficits (*Uezu et al., 2016*). Moreover, in that study, we found carbachol-induced neuronal oscillations, which are critically dependent on GABAergic inhibition (*Mann et al., 2005*), evoked epileptic form activity in InSyn1 depleted hippocampal slice culture but not in control samples. As MEA experiments revealed increase neuronal activity both at the local and network level, we speculated InSyn1 KO animals might display susceptibility to seizure-like activity. Many iPSD gene-mutations are either directly associated with epilepsy syndromes or with other

neurodevelopmental disorders that are often accompanied by seizures (*Ali Rodriguez et al., 2018*; *Fritschy, 2008*; *Uezu et al., 2016*). However, no obvious spontaneous epileptic endophenotype was observed in the InSyn1 KO mice during normal handling. Furthermore, there was no difference between WT and KO mice in the onset, severity, or duration of seizures induced by the GABA receptor antagonist Flurothyl. The lack of an overt seizure phenotype could be due to several reasons. First, it is possible that InSyn1 KO mice manifest different types of epilepsy syndromes with no convulsion seizure phenotype, such as absence epilepsy. Absence epilepsy mostly occurs among children of ages 6–15 characterized by a sudden loss of consciousness with no shaking or falling with a typical 3 Hz spike-and-wave discharges recorded by electroencephalograms. Several mice and rat models exist recapitulating similar phenotypes (*Polack et al., 2007*; *Sasaki et al., 2006*; *Tan et al., 2007*), with the underlying pathological mechanism implicated by thalamocortical circuit abnormalities (*Paz and Huguenard, 2015*). As InSyn1 is broadly expressed in multiple brain regions, EEG recordings would be required to further examine the evidence of this type of epileptic-like activity. Despite the apparent lack of overt seizure susceptibility of the InSyn1 KO mice, we did observe significantly abnormal neuronal firing in vivo by imaging neuronal activity in the dentate gyrus, a region we found to exhibit high levels of InSyn1 expression. We found a striking increase in neuronal firing rate in KO mice compared to WT littermates, demonstrating InSyn1 loss significantly disrupts normal patterns of neural activity. This was consistent with the in vitro local field recordings of InSyn1 KO neurons, which is also evidence of abnormally elevated excitation. Complex bursting activity in cultured neuronal networks depends on glutamate signaling, while GABAergic inputs participate in the termination of the neuronal bursting behavior and shape it (*Chiappalone et al., 2006*; *Cohen et al., 2008*). Indeed, we found an increase in the duration of bursting activity as well as synchronized activity at DIV14 from InSyn1 KO neurons. This is consistent with the data that InSyn1 is involved in inhibitory synaptic transmission and its expression starts from DIV8 and gradually increases. These defects seen in the later developmental stage in vitro, correlate well with the increase of in vivo neuronal activity we have observed in the dentate gyrus, a region where InSyn1 is highly expressed in the adult mouse brain. Also, it is worth noting that the DGC is crucial after neuronal development is complete, as suggested by several studies (*Früh et al., 2016*; *Pribiag et al., 2014*). Interestingly, *INSYN1* (or *C15orf59)* has recently been identified as a candidate gene for epilepsy and intellectual disability based on the genetic study of an individual harboring a 15q24.1 BP4-BP1 microdeletion (*Huynh et al., 2018*). Moreover, Duchenne muscular dystrophy (DMD) patients tend to exhibit abnormal EEG recordings and there are patients harboring mutations in DMD with cognitive deficits but without muscular dystrophy (*de Brouwer et al., 2014*; *Nakao et al., 1968*).

Based on these studies and the abnormal neuronal activity in the dentate gyrus of InSyn1 KO mice in vivo, we determined whether loss of InSyn1 impacted hippocampal-dependent behavior. While the InSyn1 KO mice were largely indistinguishable from their WT littermates in most behaviors we tested, we did observe an impairment of a specific learning and memory task. The hippocampus, especially subregions including DG, is essential for context-dependent fear memory encoding necessary for contextual freezing (*Bernier et al., 2017*; *Hernández-Rabaza et al., 2008*; *Kim and Fanselow, 1992*; *Maren, 2001*; *Maren et al., 2013*; *Xu et al., 2016*; *Zelikowsky et al., 2014*). In line with the elevated Ca$^{2+}$ transient activity observed in the open field, we found evidence of elevated neuronal activity during the contextual memory retrieval in the DG region by c-Fos staining. This elevated c-Fos staining correlated with a significant defect in contextual fear memory in the KO mice. Prior studies suggest that a subset of DG neurons is activated during fear memory acquisition and the precise reactivation of the same neurons can drive fear expression (*Liu et al., 2012*). For proper memory recall, a matched constellation of cells should be reactivated and altering this process can attenuate memory retrieval (*Denny et al., 2014*; *Ramirez et al., 2013*; *Ryan et al., 2015*). It is possible that abnormal patterns of activity in several brain regions contribute to the impaired fear memory phenotype we observe, yet because InSyn1 null neurons show abnormal patterns of elevated neuronal activity in the DG, it is tempting to speculate that trace memory ensemble coding is disrupted.

In summary, by analyzing a novel component of the iPSD, InSyn1, we have uncovered a new molecular link between InSyn1 and the DGC, in vivo neuronal activity patterns, and cognitive behavior. Collectively, these data demonstrate the importance of InSyn1 for inhibitory post-synaptic function in vitro and in vivo. Our findings also highlight how dysregulation of the DGC-containing inhibitory synapses can negatively impact normal brain activity and aspects of learning and memory.

Based on its recently described proteome, realizing how postsynaptic inhibition is regulated at the mechanistic level is still in its infancy. Our analysis of InSyn1 underscores the molecular complexity of inhibition that will be critical to further unravel to better understand GABAergic control of neural network function and how it may relate to human neurological disorders.

# Materials and methods

**Key resources table**

| Reagent type (species) or resource | Designation | Source or reference | Identifiers | Additional information |
|---|---|---|---|---|
| Genetic reagent (*M. musculus*) | Rosa26-LSL-Cas9 knockin | Jackson Laboratory | IMSR Cat# JAX:026175, RRID:IMSR_JAX:026175 | |
| Genetic reagent (*M. musculus*) | H11-CAG-Cas9 knockin | Jackson Laboratory | IMSR Cat# JAX:028239, RRID:IMSR_JAX:028239 | |
| Cell line (*H. sapiens*) | HEK293T | American Type Culture Collection | ATCC Cat# CRL-3216, RRID:CVCL_0063 | |
| Antibody | rat monoclonal anti-HA | Roche | Roche Cat# 3F10, RRID:AB_2314622 | ICC 1:1000 |
| Antibody | mouse monoclonal anti-alpha-Dystroglycan | Millipore | Millipore Cat# 05–593, RRID: AB_309828 | ICC 1:200 |
| Antibody | mouse monoclonal anti-HA | Covance | Cat# AFC-101P-1000, RRID:AB_291231 | WB 1:1000 |
| Antibody | rabbit polyclonal anti-GFP | Molecular Probes | Cat# A-11122, RRID:AB_221569 | WB 1:1000 |
| Antibody | mouse monoclonal anti-Gephyrin | Synaptic Systems | Cat# 147 011, RRID:AB_887717 | ICC 1:300 |
| Antibody | rabbit polyclonal anti-MAP2 | Synaptic Systems | Cat# 188 002, RRID:AB_2138183 | ICC 1:1000 |
| Antibody | mouse monoclonal anti-beta-Tubulin III | Sigma-Aldrich | Cat# T8660, RRID:AB_477590 | ICC 1:1000 |
| Antibody | rabbit polyclonal anti-GABAARα1 | Synaptic Systems | Cat# 224 203, RRID:AB_223218 | ICC 1:500 |
| Antibody | rabbit polyclonal anti-GABAARα2 | Synaptic Systems | Cat# 224 103, RRID:AB_2108839 | ICC 1:500 |
| Antibody | rabbit polyclonal anti-GABAARβ3 | Synaptic Systems | Cat# 224 403, RRID:AB_2619935 | ICC 1:500 |
| Antibody | rabbit polyclonal anti-GABAARγ4 | Synaptic Systems | Cat# 224 003, RRID:AB_2263066 | ICC 1:500 |
| Antibody | rabbit polyclonal anti-Vgat | Synaptic Systems | Cat# 131 002, RRID:AB_887871 | ICC 1:1000 |
| Antibody | mouse monoclonal anti-dystrophin | Abcam | Cat# ab7164, RRID:AB_305740 | ICC 1:100 |
| Antibody | rabbit polyclonal anti-c-Fos | Millipore | Cat# ABE457, RRID:AB_2631318 | IHC 1:5000 |
| Recombinant DNA reagent | pSpCas9(BB)—2A-GFP (PX458) | Addgene | RRID: Addgene_48138 | |
| Recombinant DNA reagent | pBetaActin-HA-α 1-syntrophin | This paper | | See Materials and methods section |

*Continued on next page*

*Continued*

| Reagent type (species) or resource | Designation | Source or reference | Identifiers | Additional information |
|---|---|---|---|---|
| Recombinant DNA reagent | pBetaActin-HA-β-dystrobrevin | This paper | | See Materials and methods section |
| Recombinant DNA reagent | pAAV-U6-sgRNA-hSyn-Cre | PMID: 27609886 | | backbone of AAV CRISPR constructs |
| Recombinant DNA reagent | pAAV-U6-InSyn1Cterm-HITI-smFP-SynI-Cre | This paper | | See Materials and methods section |
| Recombinant DNA reagent | pAAV-mTubb3 | Addgene | RRID: Addgene_87116 | HITI construct |
| Recombinant DNA reagent | pAAV-hSyn-hChR2(H134R)-EYFP | Addgene | RRID: Addgene_26973 | |
| Sequenced-based reagent | gRNAs | This paper | | See Materials and methods section |
| Chemical compound, drug | InFusion cloning kit | TaKaRa | Cat#638910 | |
| Chemical compound, drug | Bis(2,2,2-trifluoroethyl) ether/Flurothyl | Santa Cruz Bioteh | 333-36-8 | |
| Software, algorithm | GraphPad Prism | GraphPad Prism (https://graphpad.com) | RRID: SCR_002798 | Version 8 |
| Software, algorithm | SPSS | IBM | RRID: SCR_002865 | |
| Software, algorithm | ImageJ | ImageJ (http://imagej.nih.gov/ij/) | RRID: SCR_001935 | 1.52e |
| Software, algorithm | Puncta Analyzer/ImageJ | PMID: 21113117 | | 1.29 |

## Animals

C57BL/6J (stock no. 000664), Rosa26-LSL-Cas9 knockin (stock No. 026175), and H11-CAG-Cas9 knockin (stock #028239) mice were purchased from Jackson Laboratory. All mice were housed (3–5 mice per cage) in the Duke University's Division of Laboratory Animal Resources facilities. All procedures were conducted with a protocol approved by the Duke University Institutional Animal Care and Use Committee in accordance with the US National Institutes of Health guidelines.

## InSyn1 null mice

InSyn1 KO mice were generated by CRISPR/Cas9 gene-editing method at the Duke Transgenic Core Facility (Durham, NC). Briefly, *Insyn1* oligo sequence (ATGGTCATCGGGCAACTTGA) was cloned into px458 (Addgene; 48138). sgRNAs were transcribed in vitro by MegaShortScript T7 kit (Invitrogen). *Insyn1* sgRNA was microinjected into an oocyte from C57BL/6J strain mice along with Cas9 RNA with standard protocol and embryo was transfer into the pseudopregnant female mice. Chimeric offspring were bred to C57BL/6J, establishing germ-line transmission and further bred to expand the colony. Primer oligos F1 (GGGCCGTTAAAATGTGGAGC), R1 (CTCTCAGGATGCCCGATG), and R5 (TTCTCTCAGGATGCCTTCAAGT) were used for genotyping.

## Plasmids

HA-gephyrin, HA-α1-syntrophin (BC026215) and HA-β-dystrobrevin (BC016655) expression constructs were made by subcloning each cDNA along with the epitope tag into the pBetaActin plasmid (*Uezu et al., 2016*). AAV expressing InSyn1-HA and InSyn1-GFP deletion mutants were generated by PCR based In-Fusion cloning (Clontech) into the AAV backbone using mouse InSyn1 as the template or synthesized as a gBlock (Integrated DNA Technologies, Coralville, IA) to insert an HA tag or

to delete the regions encoding a.a. 1–60 (InSyn1ΔN), a.a. 61–87 (InSyn1ΔM1), a.a. 88–180 (InSyn1ΔM2), or a.a 181–292 (InSyn1ΔC). AAV CRISPR constructs were made by cloning each gRNA into pAAV-U6-sgRNA-hSyn-Cre (*Uezu et al., 2016*), (mouse *Dag1*; ACCGTGGTTGGCATTCCA-GACGG, mouse *Gphn*; GTTGGTGAGGATCATTCCCTCGG). To label endogenous mouse InSyn1, AAV HITI construct (pAAV-U6-InSyn1Cterm-HITI-smFP-SynI-Cre) was generated by cloning smFP-HA flanked by sgRNA (AAGCTAAGGGCAAGAACTAGGGG) targeting the C terminus of mouse *Insyn1* into pAAV-U6-sgRNA-hSyn-Cre (*Suzuki et al., 2016*). pAAV-mTubb3 was a gift from Juan Belmonte (Addgene # 87116) and pAAV-hSyn-hChR2(H134R)-EYFP was a gift from Karl Deisseroth (Addgene # 26973). All constructs generated in the lab were validated with sequencing (Eton).

## AAV production and neuronal infection

Large-scale AAV productions were performed as previously described (*Uezu et al., 2016*). Briefly, HEK293T cells were cultured in DMEM containing 10% fetal bovine. $1.5 \times 10^7$ HEK293T cells per 15 cm dish were plated 1 day before transfection for a total of six dishes per virus. Next day, cells were transfected with 30 μg pAd-DeltaF6, 15 μg serotype plasmid AAV2/9, and 15 μg AAV plasmid carrying the transgene with PEI MAX (Polysciences 24765). Eighteen hours later, the medium was replaced with 20 ml DMEM + 10% FBS. 48 hr later, cells were collected and centrifuged at 1200 rpm for 5 min at room temperature. The final cell pellet was resuspended in cell lysis buffer (15 mM NaCl, 5 mM Tris-HCl, pH 8.5) and freeze-thawed three times. The cell lysate was treated with 50 U/ml of Benzonase at 37°C for 30 min and centrifuged at 4500 rpm for 30 min at 4°C. The supernatant was added over a gradient of 15%, 25%, 40% and 60% iodixanol solution and centrifuged using a Beckman Ti-70 rotor, spun at 67,000 rpm for 1 hr. The viral solution was washed with 1X PBS three times and concentrated to 200 μl using a 100 kDa filter (Amicon). Aliquots were stored at −80°C until use. Small-scale AAV production followed the recently published method (*Gao et al., 2019*). In brief, HEK293T cells were plated on a 12-well plate, then transfected with 0.4 μg AAV plasmid, 0.8 μg helper plasmid pAd-DeltaF6, and 0.4 μg serotype 2/1 plasmid per well with PEI Max when cell density reached 60–80% confluency. Cells were then incubated in glutamine-free DMEM (Thermo-Fisher 11960044) supplemented with 1% Glutamax (ThermoFisher 35050061) and 10% FBS for 3 days. The AAV-containing supernatant medium was collected and filtered through a 0.45 μm filter tube and temporarily stored at 4°C. HEK293T cell line was purchased from the Duke Cell Culture Facility, which tests for mycoplasma contamination.

## Primary neuronal culture and immunocytochemistry

Primary neuronal cultures from mouse hippocampus were prepared as described previously (*Carlson et al., 2011*). Briefly, P0 pups were rapidly decapitated and hippocampal neurons were collected and plated onto poly-L-lysine-coated coverslips. Neurons were cultured in Neurobasal-A medium with 2% (v/v) B-27 supplement and 1% (v/v) GlutaMAX. Samples were fixed at DIV13-14 in 4% PFA/4%sucrose in PBS for 10 min at 37°C. They were permeabilized and blocked with 0.2% Triton X-100% and 5% normal goat serum in 1x PBS at room temperature. Samples were then incubated for 1.5 hr at room temperature with primary antibodies: rat anti-HA (Roche, 3F10; 1:1000), mouse anti-α-dystroglycan (Millipore IIH6C4; 1:200), mouse anti-gephyrin (Synaptic Systems 147011 1:300), rabbit anti-MAP2 (Synaptic Systems 188002 1:1000), mouse anti-β-III tubulin/Tuj-1 (Sigma T8660 1:1000), chicken anti-β-III tubulin/Tuj-1 (Synaptic Systems 302306 1:1000), rabbit anti-GABA$_A$Rα1 (Synaptic Systems 224203 1:500), α2 (Synaptic Systems 224103 1:500), β3 (Synaptic Systems 224403 1:500), γ2 (Synaptic Systems 224003 1:500), rabbit anti-Vgat (Synaptic systems 131002; 1:1000), mouse anti-dystrophin (Abcam MANDRA1 1:100). After washing with 1x PBS, samples were incubated with species-specific secondary antibodies in a combination of Alexa Fluor 488, Alexa Fluor 568, Alexa Fluor 647 (Invitrogen, 1:500), Alexa Fluor 405 (1:100), or DAPI (Sigma D9542; 1:1000) staining for 30 min at RT. Samples were washed with PBS and mounted with mounting media (FluorSave Reagent, EMD Millipore, 345789).

## Quantification of endogenous or exogenous protein puncta in cultured neurons

Images were taken by either Zeiss Axio Imager M2 epifluorescence microscope or Zeiss LSM 710/880 inverted confocal microscopes. All images were acquired blinded to the experimental

conditions. All the parameters taken by the microscopes were consistent between each sample. Each experimental condition was represented by 3–5 coverslips per experiment, and 5–6 images were captured per coverslip. ImageJ 1.52e (NIH) was used for image analysis, applying identical analysis parameters across all conditions within each experiment. To quantify the endogenous gephyrin, αDG, or dystrophin puncta density, a mask image was generated from the GFP fill-in signal to extract fluorescently-labeled puncta. Puncta Analyzer plugin for ImageJ 1.29 or Analyze Particles was used to count the puncta number per image (*Ippolito and Eroglu, 2010*). Background intensity was subtracted from the images (rolling ball radius = 50) and threshold to detect discrete puncta size between 0.2 and 1.0 $\mu m^2$. The density was calculated by dividing the puncta number with the area of GFP fill-in. To quantify the expression profile of InSyn1 and its mutants in neurons, we calculated the average absolute deviation of GFP or HA-positive puncta images. ROI dendritic segments, 10 um away from the cell body, were chosen from InSyn1-GFP or InSyn1-HA positive neurons. A segmented line of 15 um was drawn over the dendrite to export the intensity plot profile (ImageJ). Each pixel intensity was normalized to the mean value within the segment. The mean absolute deviation was calculated and expressed as a 'distribution index.' To quantify αDG or GABA$_A$Rs subunit puncta density and area from WT and InSyn1 KO neurons, each of the fluorescent cluster signals was extracted after masking Tuj-1 or MAP2-positive area. These puncta images were further divided into two regions, perisomatic region within a diameter of 30 μm from the center of the soma, and neurite regions within a diameter between 30 and 100 um. Images were thresholded to detect discrete puncta. The puncta density was calculated by dividing the puncta number with Tuj-1 or MAP2-positive area. Co-stained Vgat density was analyzed simultaneously. Because of the neuronal heterogeneity of the in vitro hippocampal culture, images were captured from αDG-positive neurons (*Benson et al., 1994*).

## Immunoprecipitation experiment

Immunoprecipitation experiment using HEK293T cells were performed as previously described (*Uezu et al., 2012*). Briefly, Transfected cells were lysed with lysis buffer [25mMHepes (pH7.4), 150mMNaCl, 1mMEDTA, 1% NonidetP-40, 5mMNaF, 1mMorthovanadate, 1mMAEBSF, 2 μg/mL leupeptin/pepstatin]. The lysate was centrifuged and the supernatant was incubated with GFP trap-agarose (ChromoTek). Beads were washed with lysis buffer and the sample buffer was added and subjected to immunoblotting. The protein-transferred nitrocellulose membrane was probed for mouse anti-HA (Covance HA.11 1:2000) and rabbit anti-GFP (Invitrogen A11122 1:1000) following secondary antibodies (Li-cor IRDye 680RD Goat anti-Mouse 1:10,000. IRDye 800CW Goat anti-Rabbit 1:10,000). Immunofluorescence signal was detected by Odyssey FC imager (LI-COR).

## Multi-electrode array (MEA)

Postnatal day 0 neurons from WT or InSyn1 KO cortex were plated in a 48-well MEA plate (Lumos 48, Axion Biosystems) and neuronal activity was recorded at day 8, 11 and 14 days after plating (DIV; days in vitro). Each well contains a 4 × 4 grid of 50 nm diameter electrodes with a pole-to-pole electrode spacing of 350 um. Wells were coated with 1 mg/mL of poly-L-lysine in sodium borate buffer, pH 8.5. Neurons were plated at 120,000 cells/well spotted onto the electrode grid within the inner well. At DIV5, 5 μM of AraC was added and fed every other day using basal media. Extracellular recordings of spontaneous or optogenetically induced action potentials were performed at 37°C with 5% $CO_2$ using a Maestro MEA system and AxIS software (Axion Biosystems). Ten min after the MEA plates were placed on the stage, 10 min recordings were used to calculate the metrics. Only wells that show more than 12 active electrodes were included for further data analysis. Data were acquired at a rate of 12.5 kHz filtered with a digital Butterworth bandpass of 200–3000 Hz. The threshold for spike detection was fixed at 6x standard deviation. Independent measurements were taken from triplicate MEA plates with 16–24 wells for each condition. Electrode bursting is defined as a minimal of 5 spikes separated by less than 100 ms between each spike. Network bursting is defined as a minimal of 50 spikes in a well separated by less than 100 ms between each spike with a minimum of 35% electrodes participated in the burst activity. The synchrony was examined by well-wide cross-correlogram across all unique combinations of electrodes in a well, normalized by the inter-electrode cross-correlations (*Halliday et al., 2006*). For quantification, the area under the cross-correlation histogram with the synchrony window of 20 ms was used. For optogenetic

experiments, the cultures were transduced with 0.2 µL of pAAV-hSyn-hChR2(H134R)-EYFP with a titer of 3 × 1011GC/mL at DIV1. Blue light stimulation was conducted as ten times of 5msec-on 2msec-off cycles with 1 s interval at the intensity of 0, 5, 10, 25, 50, and 75%. Evoked spike count was calculated as an average of the number of spikes from 10 stimuli in 100 ms time window after light stimulation. Evoked first spike latency was an average of time from the stimulation event to the first detected spike. Data were collected from three different MEA plates.

## Proximity ligation in situ hybridization technology (PLISH)

*Insyn1* mRNA was detected using PLISH as described previously (*Nagendran et al., 2018*). Mouse brain frozen sections were prepared using Nuclease-free grade PBS and Sucrose. The 20 µm sections were fixed with 4% methanol-free formaldehyde at room temperature for 20 min, followed by incubation with Proteinase-K (20 µg/ml, #EO0491) at 37C for 9 min. Sections were dehydrated with series of ethanol then incubated with total 1 µM of Insyn1-targeting or scramble hybridization probes at 37C for 2 hr followed by bridge and circle oligo hybridization at 37C for 1 hr. The sections were incubated with T4 DNA ligase (#M020M, New England Biolabs) at 37C for 2 hr. For rolling circular amplification (RCA), the sections were incubated with NxGen phi29 DNA polymerase (#30221, Lucigen) in hybridization chambers (#622514, Grace Bio-labs) at 37C for 16 hr. RCA products were detected using Cy5-conjugated imager oligo with incubating at 37C for 1 hr. Slices were counterstained with NeuroTrace 435/455 blue-fluorescent Nissl stain (ThermoFisher). All the probes are shown in *Figure 3—source data 1*.

## Stereotaxic injections

Adult mice (>P60) were anesthetized through inhalation of 1.5% isoflurane gas and placed in a stereotaxic frame (Kopf Instruments). Mice were administered meloxicam (~10 µL/25 g) subcutaneously before the beginning of surgery to reduce inflammation. Ethanol and betadine were applied to the skin over the skull and a vertical incision was made through the skin to expose the skull. The mouse's head angle was adjusted so that the primary fissures of the skull were on the same dorsal-ventral plane and a unilateral craniotomy was made with a high-speed drill (Foredom MH-170) over the dentate gyrus (coordinates: AP −2.0 mm; ML 1.0 mm: DV 2.0 mm), in reference to the Allen Mouse Brain Atlas (*Lein et al., 2007*). Using a precision pressure injection system (Drummond Nanoject), a glass pipette filled with virus containing a genetically encoded calcium indicator (AAV1.GCaMP6f, Inscopix Ready-to-Image virus) was lowered 200 µm below the desired depth, briefly retracted to the desired depth. After waiting 5 min, small amounts of virus were injected over a period of ~10 min (30 injections of 18–32 nL every 20 s). After waiting for an additional 5–10 min to prevent efflux of the virus during pipette retraction, the glass pipette was retracted from the brain and the skin over the craniotomy was sutured shut. After applying several drops of a topical anesthetic to the incision (bupivacaine) and administering an analgesic subcutaneously (buprenorphine,~25 µL/25 g), mice were allowed to recover under a heat lamp for 20–30 min and then placed in their home cage.

## Microendoscope imaging in freely behaving mice

Three weeks after AAV injection, a gradient index lens microendoscope (Proview Lens Probe, diameter 0.5 mm, length 6.1 mm, Inscopix [1050–002202]) was implanted in the same position during a second surgery. Briefly, a small cranial window was made above the DG and the microendoscope was slowly lowered into the brain (100 um/minute to 2.2 mm DV). After 5 min of waiting, the lens was retracted back to 2.0 mm DV and fixed to the skull using adhesive luting cement (C and B Metabond). A small screw was also affixed to the skull above the right frontal cortex to increase the stability of the implant. Cement, followed by a thin layer of tissue adhesive (VetBond) was used to seal the remaining exposed skull. One week after microendoscope implantation, a baseplate (Inscopix) was implanted over the lens, cemented in place and covered with a base plate cover (Inscopix).

5–6 weeks after initial viral injection, mice were habituated to the miniature microscope for 5–10 min before beginning the behavioral session. Calcium events were collected at 20 frames per second at 50% laser power (Inscopix nVista HD) as the animal explored an open field arena (20 cm x 20 cm) for 5 min. The onset of calcium imaging was synchronized with the placement of the animal in the arena for concurrent calcium imaging and behavioral tracking (Ethovision 11.5). At the end of each experiment, mice were perfused and localization of viral expression and lens placement were

confirmed with histology. Only mice with adequate viral expression in the dentate gyrus and correct lens placement were included in this study.

The data collected from imaging sessions was decompressed (Inscopix Image Decompressor) and processed to extract calcium event timing (Inscopix Data Processing 1.2.1). Briefly, using the Inscopix software package mentioned above, data was spatially downsampled (2-3x) and motion-corrected in reference to the first frame of the recording. Cells were identified with PCA-ICA and confirmed as cells by eye. The calcium traces extracted from PCA-ICA were then thresholded (median absolute deviation: 4; event smallest decay time: 0.20 s) and the timing of calcium events from individual cells was calculated. The timing of events was used to calculate the rate of events for each cell (Matlab). The animal's velocity during the behavioral session was extracted from Ethovision and used to define epochs of stationarity (velocity >10 cm/s for at least 1 s), after which the calcium event rate during movement epochs and stationarity epochs was calculated and normalized by the amount of time spent in each behavioral state (Matlab).

## Flurothyl-induced seizure test

We performed a vapor inhalation of flurothyl (bis(2,2,2-trifluoroethyl) ether) (ChemCruz) seizure induction in a ventilate chemical hood, testing mice individually within an air-tight glass chamber (2 L volume) from the age of P50. Mice were habituated in the chamber for 1 min before administrating 10% flurothyl in 95% ethanol, though a 10 mL syringe with 18G needle. The flurothyl solution was dripped onto a filter paper (Whatman, 1001–042, grade 1) at the top of the chamber with an infusing rate of 200 µL/mL. We video-recorded the resultant behaviors and measured the latency to the following events. Myoclonic seizure; a brief but a distinct contraction of the body and extremities but maintain postural control. Generalized seizure; convulsions with a loss of postural control. Once the mice exhibit a generalized seizure, the chamber was quickly opened to fresh air, and the mice were removed from the chamber to end the seizure. Between the trials, the chamber was cleaned with water, 70% ethanol, and replaced with a new filter paper.

## Open field test

The test was performed as described previously (*Kim et al., 2014*) from the age between P60 to P120. Mice were placed in an open field (AccuScan Instruments), and their activities were monitored over 1 hr under 350 lux illumination using VersaMax software (AccuScan Instruments; Columbus, OH). Locomotor (distance traveled), stereotypical activities (repetitive beam-breaks < 1 s), and anxiety level (duration in the center area of the arena) were measured in 5 min time-bins.

## Novel object recognition test

The test was performed as described previously (*Carlson et al., 2011*). Before testing, the mice were acclimated to the arena (48 × 22×18 cm) under 80–100 lux illumination. The test consisted of three phases, training (Train), short-term memory (STM), and long-term memory (LTM). The first day, animals were individually placed in the arena and presented with two identical objects for 10 min (Train). Mice were then stayed in their home cage for 30 min before being returned to the test arena for 10 min to assess their STM, where one training object was replaced with a novel object. Next day, the mice were re-examined 24 hr later to assess their LTM with the familiar training object paired to a second novel object for 10 min. All the tests were video recorded and analyzed by Ethovision (Noldus). Contact with a given object was defined as the mouse approaching the object with the nose being within 1 cm of the object. Preference score for the novel object was expressed as a ratio of the total time spent with the familiar object subtracted from the total time spent with the novel object, divided by the total time spent exploring on both objects.

## Morris water maze

Morris water maze task was conducted as described (*Porton et al., 2010*). A 120 cm diameter water tank was used. Opaque water in the tank was maintained at 25℃. The water pool was divided into four quadrants (NE, NW, SE, and SW). A 12 cm diameter round platform was submerged 1 cm below the water surface and 20 cm apart from the wall of the water tank at the NE quadrant. Testing consisted of three sessions: acquisition and probe trials (day 1 day 8), reversal acquisition and reversal probe trials (day 9 day 16). 1 week prior to testing, all mice were handled daily for 5 min and

then were placed in a pan of shallow water (1 cm) for 30 s to acclimate them to water. On the seventh day after handling, each mouse was placed onto the hidden platform in the NE quadrant for 20 s and then allowed to swim freely for 60 s before being returned to the platform for 15 s. Acquisition testing consisted of 32 trials given across 8 days with four trials administered per day. Trials were run in pairs, with each pair separated by 60 min. Probe trials were conducted without a platform at the end of days 2, 4, 6 and 8. Reversal acquisition and reversal probe tests were conducted the same way as the acquisition/probe tests described above, but the platform location was moved from NE to SW. For each trial, the release point for the animals was randomized across seven equally spaced points along the perimeter of the maze. All test trials were 1 min in duration. The swim time was analyzed by Ethovision (Noldus Information Technology).

### Light-dark transition test

Mice were placed in a two-chambered apparatus (Med-Associates) with a darkened (<1 lux) and a lighted chamber (~750 lux) to explore freely for 5 min. The mouse location and activity was tracked by Infrared beams.

### Contextual fear conditioning

Fear conditioning was performed as described previously (*Kim et al., 2014*; *Porton et al., 2010*). Mouse fear conditioning chamber (Med Associates, Inc) was used for conditioning and testing. Following a 2 min acclimation in the conditioning chamber, mice received a 0.4 mA scrambled foot shock for 2 s. Each mouse remained in the chamber for an additional 30 s before being placed in the home cage. Fear contextual memory was tested the next day in the same conditioning chamber for 5 min in the absence of the foot shock. Startle shock threshold was conducted with the Med-Associates startle platform (St Albans, VT) run by Startle Pro Software. Mice were placed onto a multibar grid in a Plexiglass tube which allowed the animal to move freely back and forth, but not rears upright. After 2 min of acclimation, the mouse was subjected to a total of 10 scrambles of intensities from 0 to 0.6mA separated by varying intervals of 20–90 s. Startle responses by the animal during the first 100 msec of the shock were transduced through the load cells in the holding platform. The area under the curve (AUC) of the startle reactivity was measured and expressed as arbitrary units (AU).

### c-Fos immunohistochemistry and quantification

Animals from the age between P60 to P70 were singly housed 2 days before fear conditioning to minimize the abnormal behaviors as well as c-Fos activation (*Rodriguiz and Wetsel, 2006*). After mice were perfused transcardially with 4% PFA in PBS (pH 7.4), brains were dissected, post-fixed in PFA overnight at 4C, and cryoprotected in 30% sucrose. Every third brain coronal section (40 µm) of the dorsal hippocampal regions within Bregma −1.0 to −2.0 mm was collected from each brain sample. Free-floating tissue sections were first permeabilized with 1% TritonX-100/PBS at room temperature for 3 hr. After incubated in the blocking solution of 5% horse serum, 0.2% TritonX-100 in 1x PBS for 1 hr, tissue sections were stained with primary antibodies (c-Fos rabbit, Millipore, 1:5000) overnight in the blocking solution at 4C followed by secondary antibodies (Alexa555 or Alexa647 conjugated, 1:500, LT) at RT for 2 to 3 hr at RT. DAPI staining were performed 10 min to label cell nuclei. Images were acquired by confocal microscopy (Zeiss LSM 710) at 10x with identical image settings from multiple brain section (at least four slices) with anatomically similar areas. c-Fos-positive cells were counted individually and normalized by the area of the granule cell layer. All genotypes were blinded.

### Statistical analysis

Repeats for experiments and statistical tests carried out are given in the figure legends. Experiments were replicated at least three times. Statistical analysis and plotting were performed with either GraphPad Prism 8 (GraphPad Software, CA), SPSS (IBM) or Microsoft Excel. Data were tested for normal distribution with D'Agostino and Person to determine the use of parametric (unpaired Student's t test, one-way ANOVA, two-way ANOVA) or non-parametric (Mann-Whitney, Kruskal-Wallis) tests. We confirmed necessary parametric test assumptions using the Shapiro-Wilk test (normality) and Levene's test (error variance homogeneity) when ANOVA with repeated measure was applied.

Appropriate post hoc tests were carried out in analyses with multiple comparisons and stated in figure legends. For all bar graphs, values are expressed as mean ± s.e.m. ns, not statistically significant.

## Acknowledgements

We thank J Takatoh, B Carlson for advice on image analysis, K Walder for in vitro biochemical analysis, and E Adamson for guidance on in vivo recordings. This work was supported by NIH NINDS funding (NS102456) to SHS and a Pathways to Independence award from NHLBI/NIH (R00HL127181) to PRT and PRT is a Whitehead Scholar.

## Additional information

### Funding

| Funder | Grant reference number | Author |
| --- | --- | --- |
| National Institute of Neurological Disorders and Stroke | NS102456 | Scott Soderling |
| National Heart, Lung, and Blood Institute | HL127181 | Purushothama Rao Tata |

The funders had no role in study design, data collection and interpretation, or the decision to submit the work for publication.

### Author contributions

Akiyoshi Uezu, Conceptualization, Formal analysis, Investigation, Methodology, Writing - original draft, Writing - review and editing; Erin Hisey, Formal analysis, Methodology, Writing - original draft; Yoshihiko Kobayashi, Yudong Gao, Patrick Devlin, Methodology; Tyler WA Bradshaw, Conceptualization, Writing - review and editing; Ramona Rodriguiz, Formal analysis; Purushothama Rao Tata, Resources, Methodology, Writing - review and editing; Scott Soderling, Conceptualization, Supervision, Funding acquisition, Project administration

### Author ORCIDs

Akiyoshi Uezu http://orcid.org/0000-0001-8478-4460
Yoshihiko Kobayashi http://orcid.org/0000-0001-7031-1478
Patrick Devlin http://orcid.org/0000-0002-0359-4620
Purushothama Rao Tata http://orcid.org/0000-0003-4837-0337
Scott Soderling https://orcid.org/0000-0001-7808-197X

### Ethics

Animal experimentation: All procedures were conducted with a protocol (#A224-17-09) approved by the Duke University Institutional Animal Care and Use Committee (IACUC) in accordance with the US National Institutes of Health guidelines.

### Decision letter and Author response

Decision letter https://doi.org/10.7554/eLife.50712.sa1
Author response https://doi.org/10.7554/eLife.50712.sa2

## Additional files

### Data availability

All data generated or analysed during this study are included in the manuscript and supporting files.

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
