## [Decision Letter]

**Acceptance summary:**

This paper from the Soderling laboratory follows the biochemical identification of numerous constituents present at the synaptic complex of proteins concentrated at inhibitory synaptic connections in the brain. Among the abundant and, yet, poorly studied proteins is a molecule called InSyn1. The manuscript details a characterization of the function of InSyn1. We learn the gene expression distribution, effects on synapse organization at inhibitory synapses and some basic structure/function analysis of the protein regarding synapse localization. The authors generate a new KO mouse model and examine a number of parameters including behavior, hippocampal in vivo firing rates, and network activity in neuronal cultures. The results form an exciting and impactful addition to the field of cellular and molecular neuroscience, especially as this work attempts to address how in vitro changes in inhibitory synapse structure/composition might contribute to in vivo functional changes in neuronal activity and behavior.

**Decision letter after peer review:**

Thank you for submitting your article "InSyn1 regulates GABAergic inhibition via the dystroglycan complex and is required for cognitive behaviors in mice" for consideration by *eLife*. Your article has been reviewed by three peer reviewers, including Graeme W Davis as the Reviewing Editor and Reviewer #1, and the evaluation has been overseen by Richard Aldrich as the Senior Editor.

The reviewers have discussed the reviews with one another and the Reviewing Editor has drafted this decision to help you prepare a revised submission.

Summary:

This paper from the Soderling laboratory follows the biochemical identification of many constituents of the inhibitory synaptic complex. Among the abundant and, yet, poorly studied proteins is a molecule called InSyn1. The manuscript details a fairly exhaustive characterization of the function of InSyn1. We learn the gene expression distribution, effects on synapse organization at inhibitory synapses and some basic structure/function analysis of the protein regarding synapse localization. These are all straightforward and nicely presented. The authors generate a new KO mouse model and examine a number of parameters including behavior, hippocampal in vivo firing rates, and network activity in neuronal cultures. In total, this represents a substantial amount of work and can be considered a definitive study on the role of the InSyn1 protein. In this respect, the study succeeds nicely. The results form an exciting and impactful addition to the inhibitory synapse field, especially as this work attempts to address how in vitro changes in inhibitory synapse structure/composition might contribute to in vivo functional changes in neuronal activity and behavior. There are a few concerns, but this lab is able to address them and it should be possible to do so in a short time frame required for publication in *eLife*. In total, the authors should be congratulated for a nice study that will have a broad impact for the general readership at *eLife*.

Major Concerns:

1) To strengthen the idea that InSyn1 regulates GABAAR clusters via DGC the authors should verify whether the loss of GABAAR clusters in Figure 5 is accompanied by concurrent DGC loss (and if GABAAR clusters with gephyrin are spared).

2) In line with this, how do you reconcile the different results of Insyn1 KO for DGC (Figure 4) and GABAARs (Figure 5)?

3) Figure 4 shows that InSyn1 KO causes a decrease in cluster density in neurites (no change in soma) and increased cluster area in the soma, compared with Figure 5 that shows a decrease in a2 cluster density in soma and neurites. To assist with this, area data should be included here and these differences should be addressed.

---

## [Author Response]

Major Concerns:1) To strengthen the idea that InSyn1 regulates GABAAR clusters via DGC the authors should verify whether the loss of GABAAR clusters in Figure 5 is accompanied by concurrent DGC loss (and if GABAAR clusters with gephyrin are spared).

Thank you for this excellent suggestion. We have noticed that αDG staining was positive only in a subset of hippocampal neurons. Additionally, it is known that DGC localizes in a subset of the inhibitory synapses. This was verified by a previous study co-staining hippocampal neurons with GABA_A_Rα2 subunit and dystrophin (Brunig et al., 2002). Furthermore, it has been shown that DGC exists only in a fraction of gephyrin positive synapses by co-staining with βDG and gephyrin (Levi et al., 2002). Based on this information, we sought to analyze αDG negative neurons that should mainly express gephyrin. αDG negative cells were extracted from the same images that were analyzed before and applied the same parameters to quantify GABA_A_R α1 and α2 subunits clusters in both perisomatic and neurite regions. We found the density of α2 subunit in the neurite region was decreased. This data suggest InSyn1 depletion affects GABA_A_R subunit composition also in αDG negative cells. Because it is still unclear the underlying mechanism of how DGC regulates GABA_A_Rs expression, composition, and inhibitory synaptic transmission, we decided to change the title of our manuscript to “Essential role for InSyn1 in dystroglycan complex integrity and cognitive behaviors in mice”, focusing on the functional interaction between InSyn1 and DGC. The new data has been shown in a new Figure 5—figure supplement 1.

2) In line with this, how do you reconcile the different results of Insyn1 KO for DGC (Figure 4) and GABAARs (Figure 5)?

We appreciate this critical comment. In this manuscript, we found a tight functional interaction between InSyn1 and DGC in hippocampal neurons. Although, our previous in vivo BioID and IP-MS experiments suggest other inhibitory synaptic molecules could associate (either directly or indirectly) with InSyn1. These proteins include collybistin, Iqsec3, and several GABA_A_R subunits such as α1 and α2. We speculate InSyn1 protein-protein interactions other than DGC might contribute to the uncorrelated results we have observed between DGC and GABA_A_Rα2 puncta density in somatic or dendrite regions. This possibility is now addressed in the Discussion section of the manuscript. The functional roles of these additional potential interactions will be the focus of future studies.

3) Figure 4 shows that InSyn1 KO causes a decrease in cluster density in neurites (no change in soma) and increased cluster area in the soma, compared with Figure 5 that shows a decrease in a2 cluster density in soma and neurites. To assist with this, area data should be included here and these differences should be addressed.

We further analyzed our previous GABA_A_R staining images to extract the change of each subunit cluster area between WT and InSyn1 KO neurons. We found a decrease in the cluster area of α2 and β3 neurite regions, an opposite result we found in the change of DG cluster area. This new data suggest DGC does not strongly correlate with the changes in GABA_A_R subunits expression and each subunit alterations seem to be interdependent. Several reports support this notion such as GABA_A_Rγ2 subunit deficient neurons do not change the DGC clustering (Brunig et al., 2002). Additionally, it is known that GABAergic synaptogenesis is intact in dystroglycan-deficient neurons (Levi et al., 2002). As we respond to the first comment, we realized we need to clarify whether GABA_A_R clustering is dependent on DGC. In this revised manuscript we emphasize the functional interplay between DGC and InSyn1 that we found. The new data has been included in a new Figure 5.